# Precision Nutrition and Gut–Brain Axis Modulation in the Prevention of Neurodegenerative Diseases

**DOI:** 10.3390/nu17193068

**Published:** 2025-09-26

**Authors:** Dilyar Tuigunov, Yuriy Sinyavskiy, Talgat Nurgozhin, Zhibek Zholdassova, Galiya Smagul, Yerzhan Omarov, Oksana Dolmatova, Ainur Yeshmanova, Indira Omarova

**Affiliations:** 1Laboratory of Food Biotechnology and Specialty Food Products, Kazakh Academy of Nutrition, Almaty 050008, Kazakhstan; dilyar117@gmail.com (D.T.); yerzhan_omarov@mail.ru (Y.O.); dolmatova1967@mail.ru (O.D.); 2School of General Medicine, Asfendiyarov Kazakh National Medical University, Almaty 050012, Kazakhstan; 3Neurosis and Alzheimer’s Disease Treatment Center, Almaty 050008, Kazakhstan; zhibek_zholdas@mail.ru; 4Department of Food Biotechnology, Faculty of Food Technology, Almaty Technological University, Almaty 050012, Kazakhstan; s.galiya_22@mail.ru; 5Institute of Innovative and Preventive Medicine, Asfendiyarov Kazakh National Medical University, Almaty 050012, Kazakhstan; a.yeshmanova@gmail.com (A.Y.); indiraomarova.kz1989@gmail.com (I.O.)

**Keywords:** neurodegenerative diseases, Alzheimer’s disease, Parkinson’s disease, gut–brain axis, dysbiosis, precision nutrition, nutrigenomics, chrononutrition

## Abstract

In the recent years, the accelerating global demographic shift toward population aging has been accompanied by a marked increase in the prevalence of neurodegenerative disorders, notably Alzheimer’s disease, Parkinson’s disease, amyotrophic lateral sclerosis, and multiple sclerosis. Among emerging approaches, dietary interventions targeting the gut–brain axis have garnered considerable attention, owing to their potential to modulate key pathogenic pathways underlying neurodegenerative processes. This review synthesizes current concepts in precision nutrition and elucidates neurohumoral, immune, and metabolic regulatory mechanisms mediated by the gut microbiota, including the roles of the vagus nerve, cytokines, short-chain fatty acids, vitamins, polyphenols, and microbial metabolites. Emerging evidence underscores that dysbiotic alterations contribute to compromised barrier integrity, the initiation and perpetuation of neuroinflammatory responses, pathological protein aggregations, and the progressive course of neurodegenerative diseases. Collectively, these insights highlight the gut microbiota as a pivotal target for the development of precision-based dietary strategies in the prevention and mitigation of neurodegenerative disorders. Particular attention is devoted to key bioactive components such as prebiotics, probiotics, psychobiotics, dietary fiber, omega-3 fatty acids, and polyphenols that critically participate in regulating the gut–brain axis. Contemporary evidence on the contribution of the gut microbiota to the pathogenesis of Alzheimer’s disease, Parkinson’s disease, and multiple sclerosis is systematically summarized. The review further discusses the prospects of applying nutrigenomics, chrononutrition, and metagenomic analysis to the development of personalized dietary strategies. The presented findings underscore the potential of integrating precision nutrition with targeted modulation of the gut–brain axis as a multifaceted approach to reducing the risk of neurodegenerative diseases and preserving cognitive health.

## 1. Introduction

Cognitive impairment represents one of the major challenges in contemporary healthcare, owing to its increasing prevalence and dependence on modifiable lifestyle and environmental factors [1]. Neurodegenerative disorders, such as Alzheimer’s disease (AD), Parkinson’s disease, amyotrophic lateral sclerosis, and multiple sclerosis, comprise a group of pathologies characterized by progressive neural loss and dysfunction of the central nervous system. Their development is driven by a complex interplay of genetic, molecular, and pathophysiological mechanisms, including pathological protein aggregation, neuroinflammation, and neurotransmitter system dysfunction, ultimately leading to irreversible decline in cognitive and motor functions [2].

Particular attention has been directed toward the potential for reducing the risk of dementia and Alzheimer’s disease in elderly adults, which has become critically important in light of the projected increase in disease burden in the near future [3,4]. In the context of global population aging, driven by increased life expectancy and declining birth rates, the prevalence of neurodegenerative diseases is expected to rise substantially in the coming decades [5]. According to the World Health Organization (WHO), by the year 2050 the number of individuals with cognitive impairment may exceed 150 million, placing a substantial burden on healthcare and social support systems [6].

Given the absence of curative treatments for neurodegenerative diseases, contemporary research has focused primarily on the development of preventive strategies. Among the most promising approaches are lifestyle modifications, particularly dietary interventions [7]. Current research suggests that dietary modifications that include antioxidants, nutrients with strong anti-inflammatory effects, and balanced eating patterns are associated with a reduced risk of dementia and Alzheimer’s disease, but the effectiveness of these dietary interventions varies depending on the source of nutrients, regional dietary patterns, and their impact on genetic and molecular regulatory mechanisms [8].

In recent years, the evidence base confirming the neuroprotective activity and preventive potential of biologically active compounds of food origin (nutraceuticals) in maintaining cognitive health and preventing the development of dementia, including Alzheimer’s disease, has expanded significantly. Natural bioactive substances such as polyphenols, carotenoids, essential fatty acids, phytosterols, essential oils, dietary fiber, pro- and prebiotics and other compounds are of particular interest due to their pronounced antioxidant and anti-inflammatory properties. In addition, these components act as active modulators of the molecular mechanisms implicated in the development of neuroinflammatory processes, including the formation of plaques and neurofibrillary tangles, which are recognized as the principal neurological hallmarks of Alzheimer’s disease [9]. Currently, their pharmacological activity, nutritional profile and potential clinical effects are the subject of intensive scientific research [10,11].

Of particular interest are dietary strategies enriched with bioactive components, which are considered as a promising non-evasive and safe approach to maintaining cognitive health and potential therapy for neurodegenerative diseases [12]. Contemporary research highlights the pivotal role of the gut microbiome—a complex ecosystem of microorganisms colonizing the gastrointestinal tract—in mediating the effects of nutritional factors on cognitive function and neuroplasticity.

Accumulating experimental data confirms the existence of a complex, bidirectional communication system between the gut microbiota and the central nervous system (CNS), exerting reciprocal regulatory effects on the physiological activity of both components [13,14]. The gut microbiota, representing the most numerous (~10^14^ microbial cells) and diverse microbial community in the human body [15], contributes to the formation of the gut–brain axis. This bidirectional neurohumoral communication system between the gastrointestinal tract and CNS operates under both physiological and pathological conditions.

Neurobiological studies have provided compelling evidence for the influence of the gut microbiota on neurogenesis, synaptic plasticity, and the shaping of behavioral patterns through gut–brain mechanisms [16,17]. These findings position the gut microbiota as a critical regulator of neuronal function and a potential contributor to the pathogenesis of various neurological disorders.

Precision nutrition aimed at modulating the gut–brain axis is a promising strategy for maintaining cognitive health and preventing neurodegenerative diseases [18]. Current research in this field focuses on the use of individual biomarkers and phenotypic characteristics to develop targeted dietary interventions that are potentially more effective than traditional nutritional recommendations [19]. Precision nutrition opens new prospects for the prevention of neurodegenerative diseases through targeted modulation of the microbiota–gut–brain axis and correction of individual epigenetic profiles taking into account the unique patient’s genetic, metabolic and microbiome characteristics [20]. This approach necessitates the development of integrated algorithms that combine nutrigenomics, metagenomic analysis, and personalized biomarker monitoring, thereby enabling a shift from generalized dietary recommendations to evidence-based, personalized interventions in clinical practice.

## 2. Methods

To ensure a targeted and high-quality review, a systematic search was conducted in the databases “Web of science”, “Scopus”, “PubMed”, “ScienceDirect”, and “Google Scholar”, using the following keywords: “Gut–brain axis”, “Cognitive health”, “Dietary Intervention”, “Gut microbiota”, “Dysbiosis”, “Neurodegenerative diseases”, “Dementia prevention”, “Alzheimer’s Disease”, “Parkinson’s Disease”, “Multiple Sclerosis”, “Microbiota Alterations”, “Precision nutrition”, “Nutrigenomics”, “Chrononutrition”, and “Metagenomics”, without restrictions on the publication date.

Inclusion criteria: the review included studies that examined the mechanisms by which the gut microbiota and its metabolites influence cognitive functions and the pathogenesis of neurodegenerative diseases, as well as research addressing the role of dietary interventions (probiotics, prebiotics, psychobiotics, dietary fibers, polyphenols, and omega-3 fatty acids) in the modulation of the “gut–brain axis”.

Selection stages. At the first stage, titles and abstracts were screened. At the second stage, full-text articles that met the relevant criteria underwent detailed analysis. The review incorporated experimental studies (in vitro and in vivo), clinical trials, as well as meta-analyses and systematic reviews aligned with the objectives of the study.

To ensure transparency and reproducibility of the analysis, the processes of literature search, selection, and critical appraisal were conducted in accordance with the PRISMA (Preferred Reporting Items for Systematic Reviews and Meta-Analyses) guidelines.

## 3. Pathophysiological Mechanisms Linking the Gut Microbiota and Neurodegeneration

The functional relationship between the central nervous system and the gastrointestinal tract is mediated by a complex neurohumoral network that enables bidirectional signal transmission [21]. A key role in this system is played by the vagus nerve (nervus vagus), which transmits peripheral signals from the gastrointestinal tract to the CNS via afferent fibers, forming the basis of the gut–brain axis [22]. This signaling axis plays a critical role in autonomic regulation, intestinal motility, and modulation of immune responses, ensuring the transduction of bioactive molecules from the intestine to the CNS [23]. This mechanism not only underscores the direct dependence of CNS function on gastrointestinal activity, but also highlights the integrative role of the vagus nerve in the neurohumoral regulation of physiological processes.

The vagus nerve is the tenth pair of cranial nerves (cranial nerve X), the longest of all cranial nerves and plays a key role in the parasympathetic division of the autonomic nervous system. The nerve originates in the medulla oblongata and innervates a number of vital organs, including the heart, lungs, spleen, and gastrointestinal tract. By transmitting neural impulses bidirectionally, the vagus nerve regulates numerous physiological processes such as heart rate, blood pressure, gastrointestinal motility and secretion, and the modulation of inflammatory responses [24].

Along with neural and hormonal signaling pathways, neurotransmitters, biologically active molecules that ensure fast and accurate synaptic communication both within the enteric nervous system and between the peripheral structures of the gastrointestinal tract and the CNS, play a special role in the functioning of the gut–brain axis [25]. The most significant neurotransmitters involved in the regulation of gastrointestinal function include serotonin (5-hydroxytryptamine, 5-HT), catecholamines (dopamine, norepinephrine, and epinephrine), γ-aminobutyric acid (GABA), and adenosine triphosphate (ATP) [26]. These neurotransmitters perform key regulatory functions in the enteric nervous system, influencing motility, secretion, sensitivity, and immune processes in the gastrointestinal tract. Disruption of their balance is associated with a wide range of diseases, and the ability of the microbiota to modulate neurotransmitter levels opens new opportunities for targeted therapy within the gut–brain axis [27].

In addition to direct neuronal and humoral mechanisms, a major modulatory system within the gut–brain axis is the hypothalamic–pituitary–adrenal system (HPA axis), which integrates stress and metabolic signals into a single regulatory circuit. This system plays a central role in the formation of neuroendocrine and behavioral responses to stress [28]. Chronic stress is accompanied by hyperactivation of the HPA axis, leading to a persistent increase in glucocorticoid (GC) levels and the development of associated pathophysiological changes [29].

Of particular interest is the modulating effect of the regulation of glucocorticoid hormone levels [30,31]. Maintaining physiological levels of glucocorticoids is a critical factor in normal neuronal development and cognitive functions, particularly the processes of memory formation and learning. These hormones carry out complex regulation of cerebral and intestinal functions through interconnected endocrine, metabolic, neuronal and immune mechanisms [32].

A key role in these processes is played by the integrity of the intestinal barrier, which acts as a selective filter: it ensures the absorption of nutrients, carries out immune surveillance, and prevents the penetration of pathogenic microorganisms [33].

Recent studies demonstrate that disruptions in the integrity of the intestinal barrier are associated not only with gastroenterological pathologies but also with a wide range of central nervous system disorders [34]. In particular, increased intestinal permeability has been shown to be linked with neurodegenerative diseases—such as dementia, Alzheimer’s disease, and Parkinson’s disease—as well as with neurodevelopmental disorders, including autism spectrum disorders, psychiatric conditions like depressive disorders, and with multiple sclerosis [35,36,37,38,39,40].

Experimental data obtained in vitro and in vivo confirm that under the influence of proinflammatory cytokines, destruction of tight junctions (TJ) and development of the inflammatory process occur even in morphologically intact areas of the intestinal mucosa [41,42].

Intestinal microbiota plays a key role in modulating the immune response through activation of inflammatory regulators toll-like receptors (TLRs) and nuclear factor kappa B (NF-κB) [43]. These reactions lead to the secretion of proinflammatory cytokines and enteroendocrine peptides (EEPs) [44]. Activation of TLR receptors by components of the intestinal microbiota triggers a cascade of intracellular signaling pathways leading to the translocation of NF-κB into the nucleus and subsequent transcription of proinflammatory cytokine genes [45]. Moreover, microbial metabolites such as short-chain fatty acids are able to modulate this process through epigenetic mechanisms, acting as an important link in the bidirectional communication along the gut–brain axis [46].

Dysbiosis of the intestinal microbiota and the associated increase in intestinal barrier permeability (“leaky gut”) play a significant role in the pathogenesis of neurodegenerative diseases such as Alzheimer’s disease, Parkinson’s disease and amyotrophic lateral sclerosis [47]. As noted above, disruption of the integrity of the intestinal epithelium promotes the penetration of proinflammatory cytokines and bacterial toxins into the systemic circulation, which leads to destabilization of the blood–brain barrier (BBB) and activation of microglia. These processes, in turn, induce the accumulation of pathological proteins such as β-amyloid and α-synuclein, which aggravates neurodegenerative changes [48].

Microbial metabolites constitute one of the key mediators in the gut–brain interaction. These metabolites perform critically important regulatory functions, participating in energy metabolism, intercellular communication, and maintaining systemic homeostasis, including modulation of the integrity of the BBB, neuronal plasticity, and neuroimmune interactions, which emphasizes their role in the pathogenesis and potential therapy of neurodegenerative diseases [49] (Figure 1).

Short-chain fatty acids (SCFAs), produced by the intestinal microbiota as a result of anaerobic fermentation of indigestible dietary fiber, are key regulatory molecules in the “microbiota gut–brain” system. Among them, acetate, propionate, and butyrate are of the greatest physiological significance.

Numerous experimental data confirm their multifunctional role in intersystem communication, realized through multiple mechanisms, including direct neuromodulatory action, indirect regulation through the immune and endocrine systems, and modulation of other signaling pathways of gut–brain communication [50,51,52]. These include the innate and adaptive immune response, metabolic and energy regulation, maintenance of the integrity of the intestinal and vascular barriers, as well as modulation of circadian rhythms, sleep and appetite [53,54].

The molecular mechanisms of action of SCFAs include activation of G protein-coupled receptors (mainly GPR43, GPR41 and GPR109A) and inhibition of histone deacetylase, which leads to epigenetic regulation of gene expression [55,56]. Of particular physiological significance is their ability to accumulate intracellularly, causing acidosis even at minor pH fluctuations [57], this change in the intracellular environment modulates calcium signaling, neurotransmitter release and functional activity gap junctions, which can potentially reduce neuronal communication and contribute to behavioral changes. The pleiotropic effects of SCFAs determine their key role as regulatory molecules in the microbiota–gut–brain interaction system [58].

Table 1 presents the major microbial metabolites and their effects on the brain.

Trimethylamine-N-oxide (TMAO) formation in the intestine is catalyzed by anaerobic and facultative anaerobic bacteria, among which the key role is played by representatives of the genera *Anaerococcus*, *Clostridium*, *Escherichia*, *Proteus*, *Providencia* and *Edwardsiella* [70].

Experimental data indicate a correlation between elevated TMAO levels and age-related cognitive dysfunction due to impaired mitochondrial function, oxidative stress, neuronal aging, and synaptic degeneration [71]. A promising area of research is the study of the possibilities of precision nutrition for microbiota-dependent modulation of TMAO levels, which may lead to the development of personalized therapeutic strategies for neurodegenerative diseases.

The microbiota plays a significant role in vitamin synthesis. Microbial vitamins synthesized by the intestinal microbiota (vitamins K, B_2_, B_9_, B_12_) are absorbed mainly in the colon and play an important role in maintaining systemic and cognitive health. Vitamin K modulates blood clotting, prevents neurodegenerative processes (Alzheimer’s and Parkinson’s diseases) and is associated with the preservation of cognitive functions in old age [72]. B vitamins participate in the prevention of neurological disorders, reduce homocysteine levels and slow down brain atrophy in cognitive disorders, which emphasizes their neuroprotective potential [73].

Taken together, these findings indicate that dysbiosis, impaired intestinal barrier function, and dysregulation of the metabolic activity of the microbiota are key factors contributing to the development of neurodegenerative diseases. The results of numerous studies highlight the potential of the microbiota and its metabolites as targets for the development of preventive and therapeutic strategies.

## 4. Dietary Interventions as Modulators of Gut–Brain Axis Mechanisms

### 4.1. Probiotics and Psychobiotics

Probiotics are live microorganisms that, when ingested in sufficient quantities, have a positive effect on the health of the host organism. It has been established that probiotics have a wide range of beneficial physiological effects, including anti-allergic action, improvement of intestinal condition (elimination of dysbiosis, strengthening of the epithelial barrier), enhancement of the immune response, reduction in symptoms of lactose intolerance, prevention of cancer, as well as a positive effect on the psychoemotional state [74].

The effect of probiotics on the pathogenesis of neurodegenerative diseases Is mediate by their interaction in the “gut–brain” system [75]. These microorganisms modulate the release of neurotransmitters by changing the species composition of the intestinal microbiota, which contributes to neuroprotective effects, including increased survival and differentiation of neurons [76]. In addition, probiotics have a regulatory effect on immune processes by controlling the synthesis of proinflammatory cytokines, changing the functional activity of dendritic cells and stimulating the differentiation of T-regulatory lymphocytes (Treg), which together helps to reduce the intensity of systemic inflammation associated with the pathogenesis of neurodegenerative diseases [77].

Particular attention is paid to psychobiotics—a subgroup of probiotics that can produce neuroactive compounds (such as GABA, serotonin, BDNF) and potentially affect cognitive functions and emotional state [78]. The mechanism of action of probiotics and psychobiotics is determined by a set of specific characteristics of these microorganisms. Key properties include their ability to maintain viability under the influence of acids, bile and pancreatic juice, which is a prerequisite for successful colonization and proliferation in the gastrointestinal tract [79].

A practical confirmation of the significance of these properties is provided by the examples of the *Lactobacillus helveticus* R0052 and *Bifidobacterium longum* R0175 strains, which demonstrate high survivability in the gastrointestinal tract (81.58% in the gastric phase). Their role as psychobiotics lies in their ability to modulate the host’s immuno-inflammatory response and microbial metabolome, which is manifested in a significant increase in the levels of anti-inflammatory cytokines (IL-10, IL-6) and SCFAs, accompanied by a simultaneous reduction in ammonia and the pro-inflammatory factor TNF-α [80].

These immunometabolic effects translate into clinically significant improvements in mental health outcomes. Clinical investigations have demonstrated that these psychobiotic strains significantly reduce symptoms of anxiety and depression, as assessed by the Hospital Anxiety and Depression Scale (HADS) and the Hopkins Symptom Checklist-90 (HSCL-90) [81]. The administration of *Lactobacillus helveticus* R0052 and *Bifidobacterium longum* R0175 has been associated with reduced stress levels and enhanced cognitive health, thereby substantiating their role in modulating the gut–brain axis and highlighting their therapeutic potential in stress-related disorders.

Some microorganisms can directly affect the vagus nerve by stimulating its motor nucleus. It is known that stress, which is a significant risk factor for the development of neurodegenerative diseases, induces hyperactivation of the HPA axis, which leads to an increase in the level of glucocorticoids, which, penetrating through the BBB, interact with specific receptors, promoting the development of neuroinflammation and oxidative damage [82].

Modern studies indicate that psychobiotics have a positive effect on the body by activating the enteric nervous system or stimulating the immune response. Their effect on psychophysiological markers of depression and anxiety has been established. The mechanisms of this effect can be realized through modulation of the HPA axis stress response, leading to a decrease in systemic inflammation, direct effects on the immune system, regulating its activity, secretion of bioactive molecules, including neurotransmitters, proteins and SFCAs [83].

It has been established that probiotic strains of the genera *Bifidobacterium* and *Lactobacillus* help to maintain the immune status and intestinal homeostasis, have a neuroprotective effect by suppressing neuroinflammatory processes and demonstrate a positive effect on cognitive functions [84]. In particular, experimental data indicate that the *Bifidobacterium breve* A1 strain causes a decrease in the expression of immunoreactive genes and a decrease in the concentration of proinflammatory markers in the hippocampus in mouse models of Alzheimer’s disease [85]. Clinical observations suggest the presence of a similar neuroprotective effect in elderly people with cognitive impairment [86].

In addition, these strains demonstrate an effect on neurodegenerative processes induced by the introduction of Aβ. After probiotic therapy using *Lactobacillus* and *Bifidobacterium*, animals showed restoration of synaptic plasticity in the hippocampus, as well as normalization of the long-term potentiation mechanism, which contributed to the improvement of neuronal transmission [87]. In addition, a decrease in the expression of microglial activation markers was noted against the background of an increase in the level of BDNF and synapsin. The obtained data indicates a potential role of these strains in the modulation of cognitive functions and spatial learning.

A number of clinical studies, despite the variability of results, also recorded positive effects of probiotics, including *Lactobacillus* and *Bifidobacterium* strains, in cognitive assessment, a decrease in the levels of proinflammatory cytokines, markers of oxidative damage (malondialdehyde, 8-hydroxy-2′-deoxyguanosine), as well as an improvement in metabolic parameters in patients with Alzheimer’s disease [88,89,90].

The presented data indicates a pronounced neuroprotective effect of probiotics and psychobiotics, manifested in a decrease in cognitive impairment and correction of dysbiotic changes. It is assumed that the observed effects may be associated with the modulation of oxidative and inflammatory processes. However, further large-scale clinical studies are needed to confirm the therapeutic potential of probiotics in the treatment of neurodegenerative diseases.

According to the consensus statements of the International Scientific Association for Probiotics and Prebiotics (ISAPP), probiotics are regarded as evidence-based agents that support human health by modulating the gut microbiome, including the prevention of inflammatory and metabolic disorders. Although direct clinical data on the prevention of neurodegenerative diseases remain limited, international experts emphasize the necessity of further research in this field and highlight the promising potential of probiotics for maintaining cognitive health in elderly patients.

### 4.2. Prebiotics and Dietary Fibers

Dietary fiber, including prebiotics, plays a key role in modulating the composition and metabolic activity of the gut microbiota, mainly through fermentation to produce SFCAs, especially butyrate, which has systemic anti-inflammatory and immunomodulatory effects (Figure 2). Consumptions of these bioactive components promotes the abundance of beneficial bacteria such as *Bifidobacterium* and *Lactobacillus*, and indirectly influences the balance of neurotransmitters (GABA and serotonin) and the integrity of the intestinal barrier by reducing permeability. Current evidence indicates an association between prebiotic and dietary fiber intake and improved cognitive function, as well as reduced anxiety and depression, highlighting their potential as a dietary component for maintaining mental health [91,92].

Dietary fiber refers to heterogenous group of non-digestible carbohydrates that can modulate the gut–brain axis through both microbiome-dependent and microbiome-independent mechanisms [93]. The most studied and significant mechanism of action is the microbiome-dependent pathway, in which fermentable dietary fibers such as inulin, fructo-oligosaccharides (FOS), and galactooligosaccharides (GOS) stimulate the growth of probiotic strains and increase the synthesis of SFCAs—acetate, propionate, and butyrate [94].

In animal experiments, GOS were found to increase the expression of brain-derived neurotrophic factor (BDNF) in the hippocampus, which is important for neurogenesis in adults [95]. Increased BDNF levels are associated with improved cognitive function and reduced severity of depressive symptoms, which has been confirmed in both experimental and clinical studies. In addition, it has been established that taking the prebiotic 2′-fucosyllactose (an oligosaccharide from breast milk) for five weeks in rats leads to accelerated acquisition of operant learning skills compared to the control group, indicating a potential improvement in neurocognitive processes [96].

Clinical studies indicate that the use of GOS is associated with a decrease in cortisol levels upon awakening and a modification of patterns of unconscious processing of emotional stimuli [97]. At the same time, in individuals with high personal anxiety, a statistically significant decrease in subjectively assessed anxiety is noted, while in healthy individuals without clinically pronounced symptoms, the effects are less pronounced. In individuals with functional disorders, such as irritable bowel syndrome, taking GOS is accompanied by an increase in the number of *Bifidobacteria* and a decrease in anxiety, while in patients with clinical depression, convincing data on the effect of prebiotics on depressive symptoms has not been obtained [98].

Despite the growing body of evidence indicating the beneficial effects of GOS on psychophysiological health in clinical populations, their impact on healthy individuals remains insufficiently studied and somewhat inconsistent. Results from a double-blind, placebo-controlled trial demonstrated that a 28-day intake of GOS did not lead to a statistically significant reduction in trait anxiety among healthy women. However, it was associated with a transient increase in *Bifidobacterium* abundance within the gut microbiota, as well as modulation of neurochemical markers (GABA) in specific brain regions [99]. These findings suggest a limited yet discernible influence of GOS on the gut–brain axis, particularly reflected in trends toward neurochemical and behavioral changes in subpopulations characterized by elevated baseline anxiety.

In addition to elevating neurobehavioral effects, another study investigated the potential ability of GOS to modulate eating behavior and macronutrient intake, an aspect of prebiotic influence on the gut–brain axis that remains less extensively studied. The results demonstrated that GOS consumption was associated with significant alterations in macronutrient intake, specifically a reduction in carbohydrate and sugar consumption accompanied by an increase in fat intake [100]. A correlational relationship was established between the GOS-induced increase in *Bifidobacterium* abundance and the observed shifts in dietary preferences, suggesting an indirect influence of the prebiotic on diet through modulation of gut microbiota consumption.

Given the pivotal role of dosage in determining the efficacy of prebiotic interventions, an important subsequent task is to establish the minimal effective dose of GOS capable of inducing significant modulation of the gut microbiome. Findings from a double-blind interventional study demonstrated that a three-week intake of low doses of GOS (1.3 and 2.0 g per day) led to a significant increase in the relative abundance of *Bifidobacterium* in the gut microbiota of healthy women [101]. Moreover, supplementation with 2.0 g/day was associated with statistically significant alterations in the overall microbial composition, with the intervention’s efficacy correlating with participants’ baseline microbial profiles.

In addition to microbiome-dependent mechanisms of gut–brain axis modulation, the effect of dietary fiber on brain function mediated by reduction in systemic inflammation, modulation of immune activity, and changes in metabolic processes, including regulation of glucose and lipid levels, is of considerable interest [93]. Thus, dietary fiber and prebiotics represent a promising tool for use in precision nutrition aimed ate modulating the gut–brain axis and maintaining cognitive health, but further randomized controlled trials are needed taking into account the type of fiber, dosage, duration of intervention, and initial status of the gut microbiota.

In its scientific opinions prepared within the framework of evaluating health claims of food ingredients, the European Food Safety Authority (EFSA) consistently emphasizes the importance of prebiotics and dietary fibers in maintaining normal intestinal function, as well as in regulating lipid and carbohydrate metabolism. Although EFSA traditionally refrains from making direct claims regarding the relationship of these effects with cognitive functions, experts note that the beneficial influence on gut microbiota composition and immune status can be considered an integral component of a comprehensive strategy for the prevention of various diseases, including cognitive disorders.

### 4.3. Polyphenols and Antioxidant

Polyphenols, with their pronounced antioxidant and anti-inflammatory properties, play an important role on modulating the gut–brain axis, which opens up new prospects for the prevention and treatment of neurodegenerative diseases. Consumption of these bioactive compounds is associated with an increase in microbial diversity, an increase in populations of beneficial bacteria (e.g., *Bifidobacterium*, *Akkermansia*, *Lactobacillus*) and a decrease in the number of pathogenic microorganisms, which helps restore the potential barrier, reduce permeability and reduce the level of systemic inflammation [102].

Dietary polyphenols, when ingested, undergo bacterial transformation with the formation of bioavailable metabolites that can affect neuroinflammation, oxidative stress and neuroplasticity. In particular, therapeutically significant polyphenols, including enterolactone and enterodiol, are secondary metabolites of lignans—non-flavonoid polyphenolic compounds abundant in plant-derived foods—produced through the metabolism of the intestinal microbiota [103]. These metabolites demonstrate acetylcholinesterase-inhibitory activity, positioning them as promising candidates for the treatment of Alzheimer’s disease and other neurological disorders [104].

Furthermore, dietary polyphenols including resveratrol and quercetin have been shown to modulate gut–brain axis activity by reducing corticotropin-releasing factor (CRF) levels, inflammation, and microbiota imbalance, which is accompanied by improvements in behavioral and cognitive performance in experimental models [105].

Microbial enzymes such as β-glucosidases produced by representatives of the genera *Bifidobacterium* spp. and *Lactobacillus* spp. Play a key role in the bacterial transformation of polyphenols and antioxidants [106]. These enzymes effectively catalyze the hydrolysis of flavonoid glycosides, promoting their activation.

As a result of enzymatic transformations, the intestinal microbiota is able to metabolize flavonoids and other antioxidants into a wide range of compounds, including glycosides, glucuronides, sulfates, amides, esters and lactones, characterized by increased bioavailability [107]. These bioactive components exhibit pronounced antihyperglycemic, anti-inflammatory and neuroprotective properties [108]. The process of polyphenol metabolism is carried out with the participation of various microbial populations and is largely determined by the composition of the intestinal microbiota, which affects the final profile of metabolites and related compounds [109].

According to modern concepts, polyphenols and antioxidants of microbial origin in combination with nanotechnological approaches that provide targeted delivery across the BBB have significant therapeutic potential in the treatment of various neurological diseases, including traumatic brain injury, Alzheimer’s disease and Parkinson’s disease.

### 4.4. Omega-3 Fatty Acids

Current data highlight the critical role of omega-3 polyunsaturated fatty acids (PUFA), in particular eicosapentaenoic (EPA) and docosahexaenoic acid (DHA), in the regulation of physiological processes, including their participation in the functioning of the gut–brain axis [110]. As key structural components of phospholipids in cell membranes, these acids help optimize their fluidity and stability. This property is essential for neuronal signaling, since it determines the efficiency of neurotransmitter biding, the speed of impulse conduction, and the integrity of intracellular signaling [111,112].

Omega-3 fatty acids demonstrate a pronounced ability to modulate the quantitative and qualitative composition of the intestinal microbiome, while the microbiota itself if actively involved in the processes of absorption and metabolic transformation of these compounds. Studies in this area have revealed a number of consistent changes in the composition of the intestinal microbiota under the influence of omega-3 fatty acids. In particular, a significant decrease in the number of representatives of the genus *Faecalibacterium* was recorded, which in some cases was combined with an increase in the proportion of bacteria of the *Bacteroidetes* type, as well as butyrate-producing microorganisms belonging to the *Lachnospiraceae* family [113].

Experimental data on animal models demonstrate that the interaction between the intestinal microbiota, omega-3 fatty acids and the immune system can play a key role in maintaining the integrity of the intestinal barrier and modulating the activity of immunocompetent cells [114]. In addition, both preclinical and clinical studies have confirmed the ability of these bioactive compounds to modulate the gut–brain axis by inducing changes in the composition of the intestinal microbiota [115,116,117].

Intestinal microbiota is able to modulate the absorption, bioavailability and biotransformation of omega-3 fatty acids both directly and indirectly [118,119]. Certain bacterial species, such as *Bacillus proteus* and *Lactobacillus plantarum*, catalyze the conversion of α-linolenic and linoleic acids into conjugated forms, linoleic (CLA) and α-linolenic (CALA) acids, which are subsequently hydrogenated to form stearic acid. This process contributes to changes in the PUFA profile in brain and myocardial tissues [120].

Despite significant progress in studying the effects of omega-3 fatty acids on the microbiota and the gut–brain axis, the mechanisms of their mutual regulation, including the effects of specific bacterial strains on PUFA metabolism, remain incompletely understood. A promising direction if the study of individual differences in the response of the microbiota to omega-3 fatty acids, which may form the basis for precision nutritional strategies. Furthermore, the long-term effects of microbiota modulation by polyunsaturated fatty acids in the context of neurodegenerative diseases require further study.

Table 2 presents summarized data from clinical studies on the effects of key bioactive component groups on regulatory mechanisms of the gut–brain axis.

## 5. Neurodegenerative Diseases and the Role of the Microbiota

### 5.1. Gut Microbiome Dysbiosis and Its Association with Dementia and Alzheimer’s Disease

In recent decades, convincing evidence has accumulated on the key role of the intestinal microbiota in regulating the function of the CNS via the gut–brain axis. It has been established that dysbiotic changes in the microbial composition are associated with the development of neuroinflammation, oxidative stress and pathological accumulation of protein aggregates, which contributes to the progression of neurodegenerative diseases, including dementia and Alzheimer’s disease (AD) [134,135]. Microbiota disturbances can lead to increased permeability of the intestinal barrier, systemic inflammation and impaired signaling along the vagus nerve, which ultimately affects neuroplasticity and cognitive functions.

Table 3 presents intestinal microbiota disturbances in Alzheimer’s disease and their relationship with the pathogenesis of neurodegeneration.

Of particular interest are data on the environment of bacterial factors in the pathogenetic cascade of neurodegenerative diseases characterized by impaired aggregation of endogenous proteins that spread via a prion-like mechanism [136,137].

In particular, the role of the enterobacteria within the concept of the “gut–brain” axis has attracted considerable attention. Experimental data demonstrate that bacterial endotoxin contributes to the aggravation of brain pathology in Alzheimer’s disease, including through the stimulation and aggregation of β-amyloid (Aβ) [138,139].

Research in this area indicates a constant expansion of the list of known amyloid systems. These structures have been identified in representatives of both Gram-positive and Gram-negative bacteria, including the genera *Bacillus*, *Pseudomonas*, *Staphylococcus*, *Streptomyces* and others, indicating a wide distribution of functional amyloids among the microbiota and their recognition through the TLR2 receptor [140,141].

In another experimental study using a transgenic mouse model of AD, age-dependent alterations in the gut microbiota composition were observed, accompanied by a decrease in the concentration of SCFAs [142]. These disturbances were correlated with the accumulation of amyloid deposits and ultrastructural damage in intestinal tissue.

As a result of the identified changes in microbial metabolism, emerging evidence indicates that the gut microbiota plays as active role in the regulation of cognitive functions, and that its imbalance is associated with the development of Alzheimer’s disease. A longitudinal study in APP/PS1 transgenic mice demonstrated that disruptions in gut microbiota composition-particularly the increased abundance of pro-inflammatory taxa such as *Escherichia-Shigella* and *Desulfovibrio* represent an early event in AD pathogenesis, preceding amyloid plaque accumulation and microglial activation [143].

Clinical studies further support the existence of substantial alterations in the composition of the gut microbiota in patients with Alzheimer’s disease. A recent study found a significant decrease in α-diversity of the microbiota, a decrease in the relative abundance of *Firmicutes* and *Bifidobacterium*, and an increase in the proportion of *Bacteroides* compared to control groups [144]. The identified dysbiotic changes were found to correlate significantly with key pathological markers of AD in cerebrospinal fluid (CSF), including a decrease in the Aβ42/Aβ40 ratio and an increase in phosphorylated tau (p-tau).

Another study found significant association between intestinal dysbiosis and AD pathology, demonstrating a decrease in the abundance of SCFA-producing bacteria (*Lachnospiraceae*, *Roseburia hominis*) and an increase in [*Clostridium*] *leptum* in patients with abnormal amyloid and p-tau levels in CSF, which also confirms the role of microbiota in the pathogenesis of neurodegenerative diseases through the modulation of neuroinflammation [145].

In another study, patients with Alzheimer’s disease showed a significant increase in the relative abundance of taxa such as *Acidobacteriota*, *Verrucomicrobiota*, *Planctomycetota*, and *Synergistota*, as well as a decrease in the content of representatives of the genera *Bifidobacterium*, *Roseburia*, *Monoglobus*, *Lactobacillaceae*, and others [146]. Differential analysis revealed an increase in the representation of the genera *Christensenellaceae* R-7 group, *Prevotella*, *Akkermansia*, and *Eubacterium*. In addition, statistically significant correlations were found between certain bacterial taxa and blood biochemical parameters, including adiponectin, bilirubin, and C-reactive protein levels.

**Table 3 nutrients-17-03068-t003:** Gut microbiota disorders in Alzheimer’s disease and their relationship with the pathogenesis of neurodegeneration.

Nature of Dysbiosis	Associated Changes in AD	Potential Mechanisms of Impact on the Central Nervous System	References
Deficiency of neuroprotective taxa	↓ *Firmicutes*, *Bifidobacterium* ↑ *Bacteroidetes*	SCFAs deficiency, activation of neuroinflammation, neurotransmitter dysregulation, disruption of the gut–brain axis	[144]
Increase in pro-inflammatory bacteria	↓ *Lactobacillus*, *Bifidobacterium*, *Ruminococcus* ↑ *Escherichia*, *Enterococcus*	Decreased SCFA production, increased lipopolysaccharide levels, immune dysregulation, development of neuroinflammation	[147]
Increase in pro-inflammatory bacteria	↓*Firmicutes*/*Bacteroidetes*, *Bifidobacterium* ↑ *Pseudomonadota*, *Synergistetes*, *Christensenellaceae*	Disrupted metabolic pathways, suppressed SCFA degradation, and sugar metabolism dysregulation exacerbate oxidative stress and promote neuronal damage	[148]
Neuro-associated dysmetabolic imbalance	↑ *Blautia*, *Enterobacteriaceae*, *Enterobacteriales*, *Gammaproteobacteria*, *Bacilli*	Neurotransmitter dysregulation, modulation of neuronal excitability	[149]
Pro-inflammatory dysbiosis with amyloid-associated microbial profile	↓ *Megamonas*, *Serratia*, *Leptotrichia*, *Clostridium* (*Clostridiaceae*) ↑ *Victivallis*, *Enterococcus*, *Mitsuokella*, *Clostridium* (*Erysipelotrichaceae*)	Disruption of the gut–brain axis through microbial translocation and systemic inflammation	[150]
Increase in pro-inflammatory bacteria	↓*Bifidobacterium* spp., *Firmicutes*, *Actinobacteria*↑ *Akkermansia*, *Enterobacteria*, *Bacteroidetes*, *Bacillus cereus*, *Prevotella*, *Clostridium* IV	Decreased SCFA production, neurotransmitter imbalance, BBB disruption, enhanced oxidative stress	[151]
Increased β-diversity	↓ *Bacteroides*, *Lachnospira*, *Ruminiclostridium_9* ↑ *Prevotella*	Reduced production of protective metabolites, enhanced systemic and neuroinflammation, progressive microbiome imbalance	[152]
Decrease in butyrate-producing bacteria and growth of opportunistic flora	↓ *Clostridiaceae*, *Lachnospiraceae* ↑ *Escherichia-Shigella*, *Bacteroides*, *Holdemanella*, *Romboutsia*, *Megamonas*	Reduced SCFA production, activation of inflammatory processes, impaired synthesis and metabolism of neuroactive compounds	[153]

↑—increase; ↓—decrease.

Significant microbial disturbances are also observed in the oral microbiome of patients with Alzheimer’s disease. And increase in overall bacterial diversity, an increase in the relative abundance of *Firmicutes*, and a simultaneous decrease in the abundance of *Bacteroidota* have been reported [154]. Additionally, significant decreases in the abundance of species such as *Haemophilus parainfluenzae*, *Prevotella melaninogenica*, *Prevotella histiola*, and *Actinomyces oris* have been revealed. These observations indicate a possible pathogenetic role of the microbial imbalances in the oral cavity in the development and progression of the disease.

Thus, intestinal and oral microbiota dysbiosis can be considered a significant risk factor for the development and progression of dementia and Alzheimer’s disease, which opens up new prospects for the development of preventive and therapeutic strategies aimed at modulating the microbial composition.

### 5.2. The Microbiota and Parkinson’s Disease: From the Gut to the Brain

Unlike the Alzheimer’s disease, where the role of the microbiota has been actively studied only in recent years, the link between intestinal dysbiosis and Parkinson’s disease (PD) was identified much earlier, due to pronounced gastrointestinal symptoms preceding motor disorders in patients. Microbiota disturbances in PD are characterized by a decrease in anti-inflammatory bacteria producing SCFAs and an increase in microorganisms associated with neuroinflammation and oxidative stress [155,156]. The main dysbiotic changes in the microbiota in Parkinson’s disease are presented in Table 4.

Parkinson’s disease is a neurodegenerative disorder characterized by α-synucleinopathy affecting all levels of the gut–brain axis, including the central, autonomic, and enteric nervous systems [165]. α-synuclein plays a central role in the pathogenesis of PD, as evidenced by its presence in Lewy bodies and neurites, characteristic morphological markers of the disease [166]. Dysbiotic changes in the microbiota typical of PD contribute to increased inflammatory responses and impaired homeostasis in the intestine. These changes are accompanied by increased permeability of the intestinal barrier and translocation of bacterial components, including lipopolysaccharides, into the systemic circulation, which can induce the expression, post-translational modifications, and aggregation of α-synuclein in neurons [167].

Considering the established link between microbial lipopolysaccharides and α-synuclein aggregation, experimental data demonstrate that chronic rotenone administration induces degeneration of tyrosine hydroxylase (TH)-positive neurons regardless of microbial status. However, the development motor deficits and increased intestinal barrier permeability were observed exclusively in conventional mice with an intact microbiota [168].

Parallel investigations in other models have revealed variability in the contribution of the microbiota to the pathogenesis of Parkinson’s disease. For instance, in SNCA A53T transgenic mice, early motor impairments, gastrointestinal dysfunction, and reduced α-diversity of the gut microbiota were observed [169]. At the same time, microbiota alterations induced by antibiotic treatment or co-housing with wild-type mice exerted only minimal modulatory effects on symptom severity.

Clinical studies have also identified significant structural alterations in the gut microbiota of patients with Parkinson’s disease. In particular, there is a decrease in the number of butyrate-producing and cellulose-degrading bacteria (*Blautia*, *Faecalibacterium*, *Ruminococcus*) against the background of an increase in the proportion of opportunistic genera, including *Escherichia-Shigella*, *Proteus*, *Enterococcus*, *Streptococcus* [170]. These changes are associated with disease progression, increase in inflammatory processes and a potential decrease in the production of short-chained fatty acids, which play a neuroprotective role.

The results of another study indicate that patients with PD have a significantly increased number of taxa such as *Verrucomicrobia*, *Mucispirullum*, *Porphyromonas*, *Lactobacillus* and *Parabacteroides*, while in the control group, bacteria of the genus *Prevotella* predominated [171]. In addition, a comparative analysis of PD subtypes revealed that patients with the non-tremor form of the disease had a more pronounced increase in the number of *Bacteroides* compared to individuals with the tremor subtype. It should be noted that the observed alterations in the microbiota were associated with increased plasma concentrations of pro-inflammatory cytokines in the patients, specifically interferon-γ (IFNγ) and TNFα. Correlation analysis revealed a positive relationship between the relative abundance of *Bacteroides* and TNFα levels, as well as between the abundance of *Verrucomicrobia* and IFNγ. These findings suggest a potential role of the gut microbiota in the pathogenesis of neuroinflammation in Parkinson’s disease.

Another study also confirmed a decrease in the number of bacterial taxa with anti-inflammatory and neuroprotective properties in patients with Parkinson’s disease. In particular, a decrease in the number of representatives of the *Lachnospiraceae* family, including SCFA-producing genera *Butyrivibrio*, *Pseudobutyrivibrio*, *Coprococcus* and *Blautia*, was found, which was accompanied by significant changes in the fecal metabolomic profile, characterized by a decrease in the concentrations of neuroprotective compounds: lipids, vitamins, amino acids and other organic metabolites [172]. The data obtained indicated a violation of metabolic homeostasis mediated by intestinal dysbiosis and emphasizes the potential of microbial and metabolic biomarkers for the diagnosis and therapy of Parkinson’s disease.

These changes are accompanied by a decrease in the phylogenetic diversity of the microbiota and significant shifts in beta-diversity, which reflects a stable restructuring of microbial communities. A recent study found a statistically significant association between the phenomenon of dopaminergic therapy wear-off and an increase in *Lactobacillus* with a decrease in *Blautia*, while in dyskinesias no significant differences in the microbiota composition were observed after taking into account confounding factors [173]. In addition, age, disease duration, the presence of motor complications, as well as lifestyle parameters (body mass index, physical activity level, diet, weight fluctuations) affect the microbiota structure and can act as confounding factors or independent modifiers.

The results of another study in this area indicate a positive correlation between the degree of the considered microbial disturbances and the severity of motor symptoms assessed by the MDS-UPDRS III scale, as well as disease progression according to the Hoehn & Yahr scale [174]. Functional metagenomic analysis revealed activation of metabolic pathways involved in lipopolysaccharide biosynthesis, as well as decreased activity of pathways for the synthesis of amino acids—precursors of neurotransmitters (including phenylamine, tyrosine and tryptophan), indicating the potential involvement of dysbiosis in disruption of neurotransmitter metabolism and the formation of a pro-inflammatory environment in the intestine.

Thus, current data confirm the key role of intestinal dysbiosis in the pathogenesis of Parkinson’s disease, mediated by disruption of the gut–brain axis through mechanisms of neuroinflammation, oxidative stress and dysregulation of α-synuclein metabolism. Further research in this direction may open new opportunities for the development of precision dietary and therapeutic strategies aimed at modulating the microbiota in order to correct neurodegenerative processes.

### 5.3. Multiple Sclerosis and Microbiota-Mediated Autoimmune Mechanism

In contrast to the neurodegenerative diseases discussed above, such as Alzheimer’s disease and Parkinson’s disease, the pathogenesis of multiple sclerosis (MS) is determined primarily by autoimmune mechanisms directed against components of the CNS, primarily myelin [175,176]. According to modern studies, myelin-reactive CD4+ T-helpers (TH-cells) play a leading role in the development of MS. Among the various subpopulations of TH-cells, TH1 and TH17 attract the greatest attention of researchers, which is due to their ability to produce proinflammatory cytokines—IFN-γ and interleukin-17 (IL-17), respectively [177]. The discovery of TH17 cells as ana independent subpopulation of CD4+ T-lymphocytes has radically changed the traditional paradigm of the TH1/TH2 dichotomy in the immunopathogenesis of autoimmune diseases. This subpopulation is characterized by a unique proinflammatory profile due to the secretion of IL-17, IL-21, IL-22 and IL-23, which contributes to the development of chronic inflammation [178]. TH17-mediated pathogenesis of multiple sclerosis includes a complex of interconnected processes: production of inflammatory cytokines (IL-17, GM-CSF), disruption of the integrity of BBB, activation of microglia and chronic recruitment of inflammatory cells of the CNS, forming a self-sustaining inflammatory cascade, which defines TH17 cells and their mediators as promising targets for targeted therapy [179].

In recent years, increasing evidence indicates that the intestinal microbiota may play a critical role in modulating the immune response and triggering the autoimmune cascade underlying multiple sclerosis [180]. According to recent data, patients with relapsing-remitting multiple sclerosis (RRMS) exhibit characteristic dysbiosis, manifested by a reduced abundance of bacteria associated with the production of SCFAs and the induction of regulatory T-cells, along with an increased prevalence of opportunistic and pro-inflammatory microorganisms [181,182]. These changes in the composition of the microbiota contribute to the activation of TH1/TH17-mediated immune responses, increased synthesis of proinflammatory cytokines (such as IL-6, TNF-α and IL-17), decreased immunoregulatory activity and increased permeability of the intestinal epithelium [183].

In an experimental study using an autoimmune encephalomyelitis model that mimics multiple sclerosis, it was established that the baseline composition of the gut microbiota determines whether the disease develops into a chronic-progressive or a relapsing -remitting form [184]. Metagenomic analysis demonstrated that immunization was accompanied by both general and phenotype-specific alterations in the microbial community, highlighting the pivotal role of the microbiota in modulating either a pro-inflammatory or a tolerogenic environment. Building on these findings regarding the role of the microbial community, the study of exogenous factors capable of inducing dysbiosis becomes increasingly important. In another experimental investigation, diet-induced obesity was shown to exacerbate the course of autoimmune encephalomyelitis in HLA-DR3 transgenic mice, leading to gut dysbiosis characterized by an increased abundance of *Proteobacteria* and *Desulfovibrionaceae*, as well as heightened intestinal permeability and systemic inflammation [185]. These results underscore the critical role of the microbiota and its associated metabolic pathways in the pathogenesis of the disease, thereby opening new perspectives for the development of targeted therapies for patients with MS and comorbid obesity.

In addition to its indirect effects through modulation of barrier function, the gut microbiota directly regulates the immune response by influencing the expression of key chemokine receptors on lymphocytes. Clinical studies have demonstrated that the gut microbiota also modulates the expression of the CCR9 receptor and the frequency of CCR9+ CD4+ T-cells, which play a crucial role in shaping both local and systemic immune responses [186]. Disruption of these mechanisms is associated with decreased immune tolerance and increased proinflammatory processes, while correction of microbial imbalance can contribute to clinical remission, which confirms the important role of microbiota in the pathogenesis of the disease.

Numerous clinical and experimental data confirm the importance of intestinal microbiota in the pathogenesis of multiple sclerosis, demonstrating stable changes in both the composition and functional activity of microbial communities in patients with this disease. Patients show a significant increase if the number of genera *Akkermansia municiphila*, *Methanobrevbacter smithii*, *Clostridium perfringens*, *Pseudomonas*, *Streptococcus*, *Dorea*, *Blautia*, which if accompanied by increased permeability of the mucous barrier, activation of innate and adaptive immunity and increased production of proinflammatory mediators [187,188,189]. At the same time, a deficiency of bacteria with immunosuppressive and barrier-protective properties is recorded, including *Faecalibacterium prausnitzii*, *Anaerostipes*, *Prevotella*, *Bacteroides*, *Parabacteroides*, *Adlercreutzia*, *Lactobacillus*, which leads to a decrease in the level of SCFAs (butyrate and propionate), impaired differentiation of regulatory T cells and an imbalance of TH17/Treg [187,190,191].

Therapeutic approaches aimed at correcting the composition of the microbiota in patients with multiple sclerosis include direct effects (the use if strain-specific probiotics), indirect effects through dietary interventions and the use of prebiotics, as well as a promising strategy of fecal transplantation [192]. Thus, intestinal microbiota is considered not only as a significant pathogenetic factor in the development of multiple sclerosis, but also as a promising target for therapy and prevention of the disease, and its modulation as a means of increasing the effectiveness of therapy and improving the quality of life of patients.

## 6. Precision Nutrition and Personalized Strategies

### 6.1. The Concept of Precision Nutrition

Precision nutrition aims to develop comprehensive and dynamic dietary recommendations that take into account the changing and interrelated parameters of the internal and external environment of an individual throughout the life cycle [193]. However, the implementation of this approach faces a number of challenges, including the high cost of technology, insufficient standardization of methods for analyzing individual biomarkers, as well as ethical and legal issues associated with the use of genetic and personal data. Figure 3 presents an integrative model of precision nutrition, outlining the main input parameters, analytical methods, and expected clinical outcomes.

The European Society for Clinical Nutrition and Metabolism (ESPEN), in its guidelines, emphasizes the importance of diets rich in antioxidants, omega-3 fatty acids, and dietary fiber for reducing the risk of cognitive impairment in older adults. The guidelines also underscore the necessity of implementing early dietary interventions as a key component of a preventive strategy.

Unlike traditional diets, precision nutrition is based on the principles of nutrigenomics and nutrigenetics, which study the interactions between food components and the molecular mechanisms of the body [194]. In nutrigenomics research, high-throughput omics technologies—including transcriptomics, proteomics, and metabolomics—are employed to analyze the impact of dietary interventions on biological systems and to investigate interactions between nutrition and genes [195]. Transcriptomics assesses changes in gene expression at the genome level and is widely used in nutrigenomics, proteomics studies the full set of proteins modulated by nutrition, and metabolomics focuses on the analysis of low-molecular-weight metabolites reflecting cellular activity [196,197,198]. Additionally, microarray technologies are used to globally analyze gene expression profiles and study transcriptional regulation under the influence of nutrients and bioactive components [199]. The integration and interpretation of data obtained using omics technologies is achieved through the development of bioinformatics, which is necessary for studying gene-nutrient interactions and developing personalized dietary strategies. The combined use of these methods allows us to identify the effects of dietary components on gene regulation, protein expression, and metabolic processes, which forms the basis of nutrigenomics and personalized nutrition. The key barrier in this area remains the interpretation of the obtained data: biological variability, the influence of external factors (e.g., stress and ecology) and the lack of universal algorithms for predicting individual responses to diet limit the transition from scientific research to clinical practice.

In this regard, within the framework of this approach, in addition to genetic characteristics, such as factors like dietary habits, eating behavior, physical activity level, microbiota composition and metabolomic profile are currently actively considered [200,201]. This comprehensive analysis is especially important in the context of the prevention of neurodegenerative diseases, since dysfunction of the gut–brain axis and chronic inflammation, modulated by dietary factors, significantly contribute to the pathogenesis of Alzheimer’s disease and Parkinson’s disease.

Modern studies demonstrate that the effectiveness of dietary interventions varies significantly among individuals, which is due to gene polymorphisms, microbiome features and metabolic status [202]. Dietary interventions have been shown to induce individual-specific changes in the gut microbiota that correlate with its baseline composition [203,204]. In this regard, the integration of gut microbiota data with other personal biological parameters represents a promising direction for predicting metabolic responses to dietary therapy.

Diet is a key exogenous factor determining the composition and functional activity of the gut microbiota, which determines its importance as a therapeutic tool for microbiome modulation [205]. Numerous intervention studies involving humans confirm that modification of macronutrient composition and inclusion of dietary supplements induce rapid and reproducible changes in the microbial profile; however, the severity and direction of these shifts demonstrate interindividual variability due to the initial α-diversity of the microbiota and phenotypic characteristics of the host [206]. In particular, high-fat and low-fiber diets are associated with reduced diversity and enrichment of pro-inflammatory taxa, whereas consumption of plant proteins and fermentable carbohydrates promotes proliferation of commensal bacteria and increased SCFA production [207,208]. These data highlight the critical role of dietary factors in regulating microbial metabolic pathways and justify the need to develop precision dietary strategies integrated analysis of the baseline microbiome, genetic potential, and dietary habits, as standardized approaches demonstrate limited efficacy due to the marked heterogeneity of individual responses.

### 6.2. Biomarkers for Nutrition Personalization

Precision nutrition is based on the identification and validation of reproducible biomarkers that can predict individual metabolic responses to dietary interventions. Recent advances in high-throughput technologies have enabled the identification of molecular signatures associated with nutrient metabolism, gut microbiota composition, and metabolic health.

Biomarkers are objective and measurable indicators that play a key role in assessing health status, identifying nutrient deficiencies, and taking into account individual metabolic characteristics. In the context of precision nutrition aimed at preventing neurodegenerative diseases, the analysis and use of biomarkers in dietary interventions helps to accurately determine the physiological needs of the body, minimizing the subjectivity of interpretation and providing evidence-based interventions [209].

Nutritional biomarkers are biochemical indicators determined in biological fluids of the body that allow an objective and specific assessment of micronutrient status. They are of particular importance in the diagnosis of micronutrient deficiency, when clinical symptoms may be absent, which excludes the possibility of using traditional screening methods on anthropometric measurements [210].

Unlike traditional approaches based on average nutrient intake rates, the use of biomarkers allows identifying specific metabolic disorders and nutrient deficiencies associated with an increased risk of neurodegenerative diseases. This opens up opportunities for targeted and safe correction, which is especially important in the context of pathogenetic mechanisms associated with oxidative stress, neuroinflammation and mitochondrial dysfunction.

Recommended nutritional biomarkers for precision therapy of neurodegenerative diseases are presented in Table 5.

The main feature of key processes that determine health status, such as lipid and carbohydrate metabolism, systemic inflammation, oxidative stress and microbiota status, if their ability to reflect different, but also interconnected aspects of metabolism [219]. These processes can be described in the form of biomarkers listed in Table 5, including metabolites and proteins. Integration of these and other biomarkers through data analysis algorithms and machine learning methods allows us to quantitatively assess the state of each of the key processes that determine cognitive health.

Modern advances in genetic testing and bioinformatics have significantly expanded the possibilities for identifying genetic biomarkers associated with predisposition to diseases, the dynamics of their progression and response to therapy [220]. Of particular importance is the development of molecular rapid tests for use in point-of-care tests (POCT). Such systems use microfluidic technologies and modern amplification methods, which ensures real-time genetic testing directly in clinical practice. This area promotes the active implementation of the principles of personalized medicine and precision nutrition [221].

Proteomic and metabolomic approaches to precision nutrition can be non-targeted (covering a wide range of biomolecules and reflecting multiple metabolic pathways) of targeted, aimed at studying specific hypothetical mechanisms linking nutritional status with cognitive health.

Recent studies in this area have identified disturbances in markers of immune function, lipid metabolism, and cellular bioenergetics that can potentially be modulated by diet [222,223]. Other studies have also found specific metabolic and proteomic signatures in the blood associated with cognitive impairment, risk of dementia, of Alzheimer’s disease biomarkers in the cerebrospinal fluid. These biomarkers are amino acids and their derivatives associated with mitochondrial function and regulation of glutamatergic transmission (branched amino acids, acyclarnites, glutamate, taurine), as well as lipids involved in maintaining the structural integrity of neuronal membranes and modulating neuroinflammation (individual subfractions of high-density lipoproteins, phospholipids, and long-chain omega-3 polyunsaturated fatty acids) [224,225,226].

Another promising direction in the prevention of neurodegenerative diseases is the use of parameter analysis using artificial intelligence (AI) in precision nutrition. This approach provides a more accurate and targeted correction of dietary factors compared to traditional universal recommendations. Machine learning methods, in particular deep learning models, allow the analysis of clinical laboratory data and potential biomarkers to predict cognitive status assessed by standardized scales as the Mini-Mental State Examination (MMSE) [227]. Such algorithms demonstrate high accuracy in identifying individual metabolic abnormalities, which makes them a valuable tool for objective quantitative monitoring of cognitive risk, as well as for assessing the effectiveness of preventive interventions.

An important advantage of artificial intelligence (AI) models based on routine biochemical indicators if their applicability for mass screening of cognitive impairment without the need for expensive neuroimaging of molecular diagnosis [228]. This opens up prospects for the widespread implementation of this approach in clinical practice, healthcare systems and preventive health insurance programs.

Nutrigenomic, epigenetic and transcriptomic approaches have significant potential for identifying the molecular mechanisms underlying precision nutrition in the context of maintaining cognitive health. However, despite the promise of these areas, their particular application currently remains limited [229].

### 6.3. Nutritional Interventions and Chrononutrition

Modern precision nutrition strategies go beyond simply taking into account the genetic and metabolic characteristics of an individual, integrating the timing of nutritional interventions to optimize their effectiveness. Chrononutrition, as a rapidly developing field, studies the influence of circadian rhythms of nutrient absorption, energy metabolism, and physiological processes associated with various diseases. This approach is based on the understanding that synchronization of dietary interventions with endogenous biological rhythms can enhance their preventive and therapeutic potential [230].

Recent studies have demonstrated that circadian regulation disorders, including desynchronosis, can initiate oxidative stress [231], neuroinflammation [232], and mitochondrial dysfunction [233]—key pathogenetic mechanisms of neurodegenerative diseases. In this regard, optimization of meal timing, macro- and micronutrients is of particular importance for the correction of metabolic disorders and maintenance of cognitive health. For example, time-restricted feeding (TRF) has been shown to improve metabolic plasticity, reduce insulin resistance, and modulate the activity of key regulators of cellular longevity and neuroprotection [234].

In addition, time-restricted eating (TRE/TRF) has a complex effect on the pathogenetic mechanisms of Alzheimer’s disease. The results of studies in this area induce that this nutritional approach helps to normalize circadian rhythms, reduce the accumulation of β-amyloid 42 (A β42), modulate anti-inflammatory cytokines, and positively change the composition of the intestinal microbiota, including an increase in its α-diversity and a decrease in the proportion of pathogenic bacteria [235]. The data obtained indicate the potential effectiveness of TRE/TRF as a preventive strategy for Alzheimer’s disease, but further studies are needed to confirm the clinical significance and clarify the molecular mechanisms (Figure 4).

Preclinical studies suggest that TRE, independent of dietary macronutrients composition, significantly improves cognitive function in aging rats compared to ad bitum-fed controls, as assessed by biconditioned associative learning tests without the influence of motivational and sensorimotor factors [236]. TRE is associated with significant changes in gut microbiota composition, including an increase in *Allobaculum* species, which correlates with cognitive performance in rats, suggesting a potential role for microbiota in modulating cognitive health during aging.

Another study in 5xFAD mice demonstrated that TRE significantly improves cognitive function, reduces β-amyloid deposition, and reduces neuroinflammation, with these effects mediated by changes in gut microbiota composition, in particular an increase in *Bifidobacterium pseudolongum* abundance and propionic acid levels [237]. The obtained results are confirmed in a clinical study, where a four-month TRE regimen resulted in a significant improvement in cognitive performance in patients with Alzheimer’s disease, with a positive correlation between fecal propionic acid levels and the degree of cognitive improvement. The neuroprotective effect of TRE is realized through the gut–brain axis by a microbiome-dependent increase in the production of propionic acids, which, penetrating the BBB, activates the FFAR3 receptor, which opens up new prospects for the development of therapeutic strategies for neurodegenerative diseases.

Intermittent fasting (IF), being a form of chrononutrition, can also significantly affect the composition and functional activity of the intestinal microbiota, which, in turn, has an indirect effect on the course of neurodegenerative diseases. A recent study demonstrated that IF induces significant restructuring of microbial communities, accompanied by an increase in α-diversity and enrichment of taxa with anti-inflammatory and neuroactive properties [238]. In particular intermittent fasting is associated with increased abundance of *Akkermansia muciniphila*, *Lactobacillus*, *Faecalibacterium prausnitzii* and *Bifidobacterium longum* bacteria capable of producing bioactive metabolites and modulating the immune response.

Another study showed that IF not only increases microbial diversity but also stimulates the growth of SCFA-producing bacteria such as *Eubacterium rectale*, *Roseburia* spp., and *Anaerostipes* spp., which are involved in the synthesis of propionate and acetate [239]. Notably, propionate, through its interaction with free fatty acid receptors (FFAR2/3), modulates vagal signaling and affects neuropeptide balance in the hypothalamus, while acetate is involved in appetite regulation and energy homeostasis [240]. In addition, intermittent fasting promotes an increase in the abundance of *Bacteroides* spp., which synthesize propionate via the succinate pathway, indicating a complex effect of chrononutrition on the metabolic profile of the microbiota [241].

Under conditions of food restriction, a metabolic shift occurs from glucose to ketone bodies, primarily β-hydroxybutyrate, which penetrates the BBB and is used by neurons as an energy source, reducing oxidative stress and stimulating the production of key neurotrophic factors [242]. The main molecular targets and effects of IF include: increased levels of BDNF, which enhances neuroplasticity, activation of SIRT3, which reduces neuroinflammation, stimulation of PGC1α, which is involved in mitochondriogenesis, an increase in GABA and ghrelin, which affect neurotransmission, induction of growth hormone and IGF-1 with neurotrophic effects, suppression of mTOR and activation of autophagy, which promotes the removal of pathological proteins and modulation of the gut microbiota, which improves cognitive function [243].

Preclinical studies in animal models indicate positive effects of intermittent fasting on cognitive functions, including improved memory performance, stimulation of neurogenesis, and reduction in neuroinflammatory processes [244,245]. Clinical studies, including observational and limited interventional studies, have shown that in elderly individuals, adherence to IF regimens, in particular limiting the time window of food intake to 8–10 h per day, is associated with improved cognitive performance and a reduced risk of developing cognitive impairment [246,247].

In addition, chrononutritive strategies involve taking into account the circadian dynamics of absorption and metabolism of bioactive compounds. The effectiveness of some neuroprotective nutrients depends on the time of their consumption, due to fluctuations in the activity of enzymes and transport systems [248]. Integrating data on circadian patterns of gene expression, hormonal profiles, and metabolic processes allows the development of personalized nutritional regimens aimed at enhancing the synergism between diet and endogenous rhythms of the body.

Another promising direction is the combination of chrononutritive interventions with real-time monitoring technologies, including wearable devices and mobile applications, which opens up new possibilities for dynamic nutritional adjustments taking into account individual circadian characteristics [249]. The introduction of these approaches into clinical practice requires further research to validate optimal nutrient intake windows and develop unified algorithms for different patients.

Thus, chrononutritive interventions represent a promising direction in precision nutrition, combining advances in nutritional science and cognitive health. Further research should be aimed at clarifying the molecular mechanisms, developing standardized protocols, and assessing the long-term effectiveness of the strategies in different patient groups.

## 7. Limitations and Future Challenges

Despite significant progress in understanding the relationship between the gut microbiota, the gut–brain axis, and the pathogenesis of neurodegenerative diseases, as well as the potential of dietary interventions, current studies remain characterized by a number of methodological and conceptual limitations.

One of the key limitations is the predominance of preclinical models, mainly in vivo animal experiments and in vitro studies, which restricts the generalizability of findings to the human population. Although numerous investigations have demonstrated the neuroprotective effects of microbial metabolites, probiotics, and bioactive compounds under experimental conditions, the evidence base derived from clinical data remains insufficiently convincing, with limited sample sizes, low statistical power, and methodological heterogeneity.

A major challenge lies in the heterogeneity of clinical studies. Existing research demonstrates considerable variability in study design, inclusion criteria, dosage regimens, intervention duration, and methods for assessing cognitive, immune and metabolic outcomes. Such variability complicates replication, hinders the development of standardized recommendations, and limits the feasibility of conducting meta-analyses necessary for an objective evaluation of the clinical efficacy of the gut–brain axis-targeted strategies.

Further complexity arises from the lack of standardized protocols for microbiota assessment, including differences in sequencing methods, sample preparation standards, and analytical approaches. These inconsistences reduce the comparability of data across studies and diminish the reproducibility of results.

At the same time, it is important to emphasize that the past decade has witnessed substantial advances in this field. In particular, the introduction of next-generation sequencing (NGS) technologies has significantly enhanced the scalability, speed, and cost-effectiveness of analyzing microbial communities associated with both the host organism and the environment. The development of high throughout analytical platforms and novel bioinformatics tools has enabled more precise characterization of the functional potential of the microbiota. Nevertheless, challenges remain regarding the standardization of protocols and the integration of these findings into clinical practice.

Additional challenges are associated with the high individual variability of the gut microbiota, which is shaped by genetic, metabolic, behavioral, and environmental factors. The uniqueness of each individual’s microbial profile limits the universality of the therapeutic solutions and underscores the need for implementing precision nutrition approaches. However, current methods for stratifying patients according to microbiome and metabolic phenotypes are still in their early stages and their clinical applicability remains limited.

Another important difficulty involves the selection of probiotic strains with confirmed activity on the CNS. Despite experimental evidence supporting the potential of psychobiotics, clinical research in this area remains scarce, and the findings are often contradictory. The lack of standardized approaches to strain selection, dosing, and administration regimens continues to hinder the development of clear clinical recommendations.

Equally significant is the heterogeneity of study populations. Factors such as age, ethnicity, dietary patterns, socioeconomic status, and comorbidities have a substantial impact on microbiota composition and the effectiveness of dietary interventions. However, these variables are not always adequately considered, which reduces external validity and complicates the systematization of research finding.

An additional limitation is the insufficient consideration of confounding factors, including lifestyle characteristics, levels of physical activity, medication use, and the psycho-emotional state of participants. Neglecting these factors reduces the accuracy of interpreting the identified associations between microbiota, dietary interventions, and cognitive impairments.

Furthermore, concerns arise from the lack of long-term prospective research for assessing the sustainability and clinical significance of nutritional interventions. Most existing studies are limited to short observation periods, making it difficult to evaluate their impact on the progression of neurodegenerative diseases, inflammatory biomarkers, and the trajectory of cognitive decline over the long term.

Taken together, these limitations underscore the need to consolidate efforts toward the development of unified methodological standards, the expansion of the evidence base, and the integration of multi-omics data (metagenomics, metabolomics, proteomics, epigenetics) into clinical algorithms. Only through the standardization of approaches, the consideration of individual variability, and the application of modern high-throughput technologies will it be possible to implement personalized nutrition- and microbiome-oriented strategies for the prevention of neurovegetative diseases.

## 8. Conclusions

The body of contemporary evidence confirms the pivotal role of the gut–brain axis in the pathogenesis of neurodegenerative diseases, including Alzheimer’s disease, Parkinson’s disease, and multiple sclerosis. Research in this field indicates that bioactive components, including prebiotics, probiotics, dietary fibers, omega-3 fatty acids, and polyphenols, modulate gut microbiota composition, immune system activity, neuroinflammatory processes, and barrier tissue function. These mechanisms may exert indirect effects on cognitive function and contribute to slowing the progression of neurodegenerative processes.

International organizations, including the WHO, EFSA, ESPEN and ISAPP, consistently emphasize the importance of functional foods, probiotics, and prebiotics in maintaining health and preventing chronic diseases. Although direct evidence regarding their impact on the risk of neurodegenerative diseases remains the subject of ongoing research, the consensus positions of these bodies support the inclusion of these components in dietary strategies aimed at preserving cognitive health in elderly.

The key approaches identified in this review include targeted modulation of gut microbiota composition through current dietary strategies, restoration of the integrity of the intestinal and blood–brain barriers, reduction in pro-inflammatory cytokine levels, and activation of neurotrophic and neuroprotective pathways mediated by microbiota-derived metabolites. In addition, particular importance is placed on the use of probiotic and psychobiotics strains capable of influencing neurotransmitter synthesis, regulating the hypothalamic–pituitary–adrenal (HPA) axis, and normalizing cognitive functions.

In this context, precision nutrition emerges as a promising and scientifically grounded strategy tailored to the individual characteristics of the patient, including microbiota composition, metabolic profile, epigenetic markers, and nutritional needs. The integration of multi-omics data into the design of personalized dietary interventions enhances the effectiveness of preventive measures and shifts the focus from treating already established pathology toward early intervention and the slowing of cognitive decline.

However, the current body of evidence is characterized by several limitations, including the fragmentary nature of clinical data, variability of results, being driven by methodological and population-specific differences, and insufficient knowledge regarding the long-term efficacy and safety of nutraceutical interventions.

The further development of this field requires the implementation of multicenter randomized clinical trials with standardized protocols; the integration of nutrigenomic, metagenomic, and metabolomic data in the design of personalized dietary strategies; as well as an in-depth analysis of the molecular mechanisms underlying the interactions between microbial metabolites, immune regulation, and cognitive functions.

Thus, precision nutrition combined with microbiota modulation should be regarded as a promising tool for preventive medicine. With further scientific validation, this approach may be integrated into comprehensive strategies aimed at reducing the risk of neurodegenerative diseases and preserving cognitive health in the population.

## Figures and Tables

**Figure 1 nutrients-17-03068-f001:**
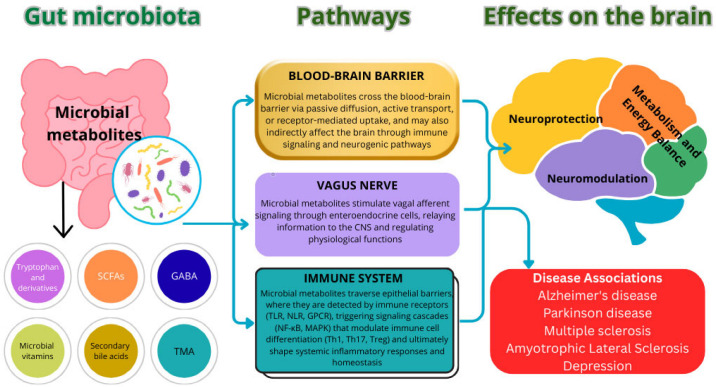
The impact of microbial metabolites on central nervous system functions within the framework of the “gut–brain” axis.

**Figure 2 nutrients-17-03068-f002:**
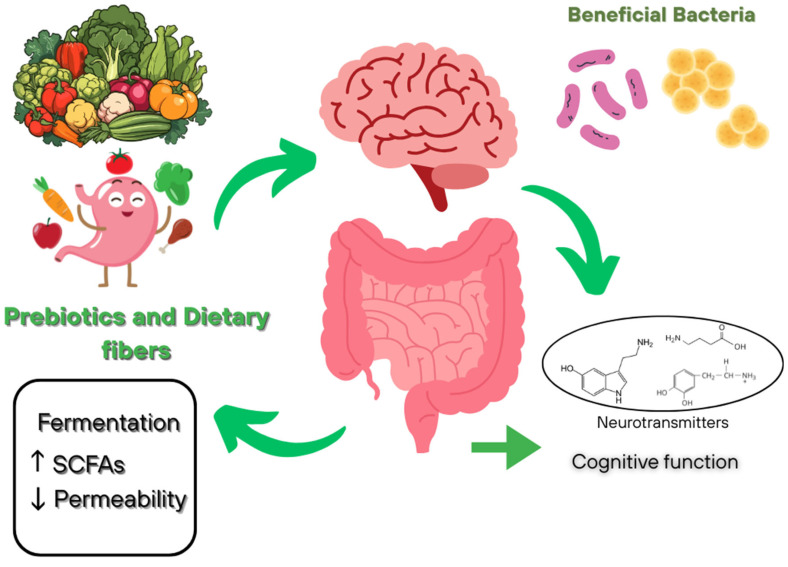
Mechanisms by which dietary fibers and prebiotics influence the gut–brain axis and cognitive health. ↑—increase; ↓—decrease.

**Figure 3 nutrients-17-03068-f003:**
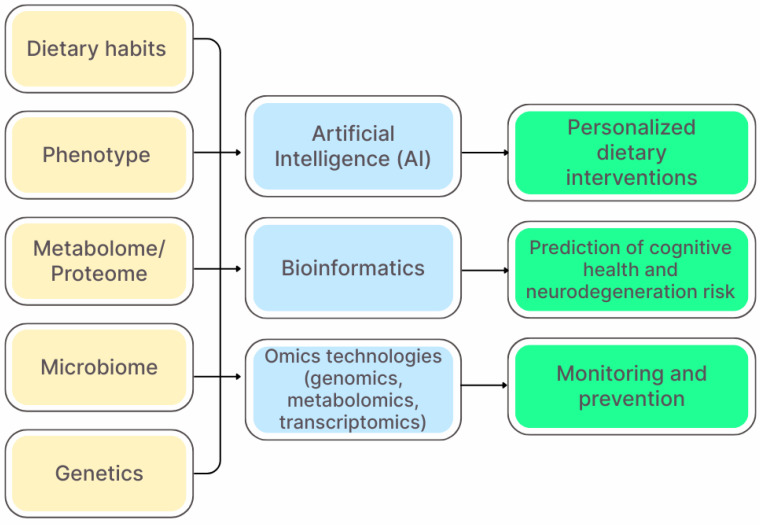
Integrative model of precision nutrition.

**Figure 4 nutrients-17-03068-f004:**
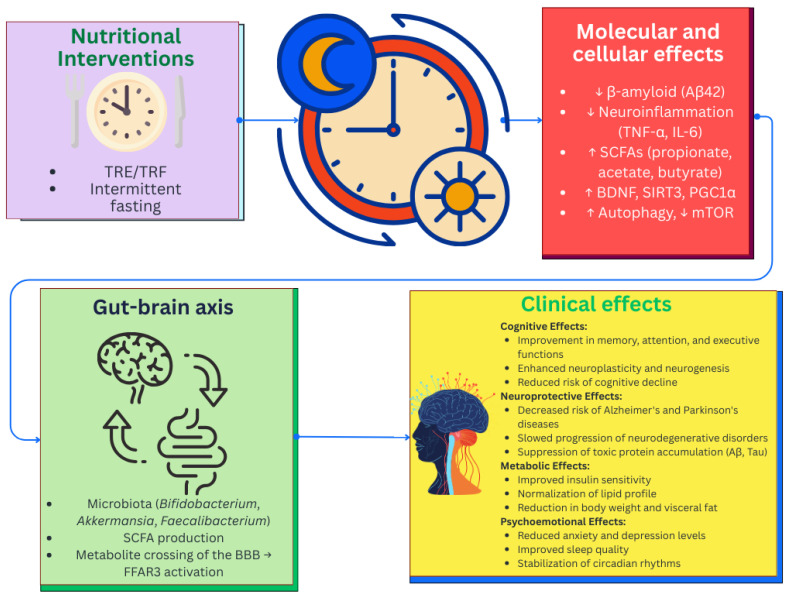
Influence of chrononutrition strategies on circadian rhythms, metabolism, gut microbiota, and cognitive health. ↑—increase; ↓—decrease.

**Table 1 nutrients-17-03068-t001:** Major microbial metabolites and their effects on the brain.

Metabolite	Primary Producers	Pathways of Influence on the Brain	Effects	Association with Diseases	References
SCFAs	*Clostridium*, *Sporanaerobacter*, *Streptococcus*, *Syntrophomonas*	BBB, vagus nerve, immune modulation	Enhanced neurogenesis, anti-inflammatory activity	Alzheimer’s disease, autism spectrum disorders, depression	[59,60,61,62]
Tryptophan and its derivatives	*Bifidobacterium*, *Lactobacillus*, *E. coli*	Gut–brain axis, neurotransmitter synthesis	Mood modulation, reduction in neuroinflammation	Depression, anxiety, schizophrenia	[63,64]
GABA	*Lactobacillus*, *Bifidobacterium*	Vagus nerve, BBB	Anxiolytic effect, reduced neuronal excitability	Anxiety disorders, epilepsy	[65,66]
Secondary bile acids	*Clostridium*, *Bacteroides*	BBB, gut–brain axis signaling pathways	Neuroprotection, regulation of apoptosis	Parkinson’s disease, multiple sclerosis	[67]
Histamine	*Lactobacillus*, *Enterococcus*	Interaction with H1–H4 receptors, effects on cerebral vasculature	Maintenance of cognitive functions, anti-inflammatory effect	Migraine, neurodegenerative diseases	[68,69]

SCFAs—Short-Chain Fatty Acids; BBB—Blood–Brain Barrier; GABA—Gamma-Aminobutyric Acid; H1–H4 receptors—Histamine receptors type 1–4.

**Table 2 nutrients-17-03068-t002:** Clinical Trial Results for Bioactive Components Modulating the Gut–Brain Axis.

Category of Component	Key Strains/Compounds	Participants	Study Design	Cognitive Effects	References
Probiotics/Psychobiotics	Bifidobacterium breve CCFM1025	Patients with major depressive disorder	Randomized placebo-controlled clinical trial	A statistically significant reduction in depressive symptoms according to the HDRS and MADRS scales, improvement in emotional state, and a decrease in the severity of psychiatric and gastrointestinal symptoms (as measured by BRPS and GSRS).	[121]
Probiotics/Psychobiotics	*Streptococcus thermophilus* NCIMB30438, *Bifidobacterium breve* NCIMB30441, *Bifidobacterium longum* NCIMB30435, *Bifidobacterium infantis* NCIMB30436, *Lactobacillus acidophilus* NCIMB30442, *Lactobacillus plantarum* NCIMB30437, *Lactobacillus paracasei* NCIMB30439, *Lactobacillus delbrueckii* subsp. *Bulgaricus* NCIMB30440	Patients with a current depressive episode	Randomized, placebo-controlled clinical trial	A significant reduction in HAM-D depression scores in the probiotic group; improvement in depressive symptoms; and decreased putamen activation in response to neutral faces (according to neuroimaging data).	[122]
Probiotics/Psychobiotics	*Lactobacillus rhamnosus GG*	Middle-aged and older adults (aged 52 to 75 years)	Double-blind, placebo-controlled, randomized trial	A significant improvement in the overall cognitive score among participants with cognitive impairment.	[123]
Probiotics/Psychobiotics	*Bifidobacterium breve* CCFM1025	Adults aged 18–65 with stress-induced insomnia	Double-blind, randomized, controlled trial	A significant improvement in sleep quality, a reduction in PSQI and AIS scores, enhanced subjective sleep quality, and a decrease in sleep disturbances.	[124]
Probiotics/Psychobiotics	*Lactobacillus acidophilus* LB	Children and adolescents aged 6–16 years with ADHD	Randomized controlled trial	A significant reduction in ADHD symptom severity compared with the control group according to the CPRS-R-L and CBCL measures, along with improvements in selective and sustained attention as assessed by the CPT.	[125]
Probiotics/Psychobiotics	*Lactobacillus rhamnosus*, *Bifidobacterium lactis*	Healthy older adults aged 55 and over	Double-blind, randomized, placebo-controlled crossover trial	Improvement in overall cognitive function, working and visuospatial memory, planning and problem-solving abilities, selective attention, cognitive flexibility, and inhibitory control, as well as a reduction in depressive symptoms and enhancement of sleep quality.	[126]
Probiotics/Psychobiotics	*Lactobacillus plantarum* PS128	Adults aged 20–40 years with chronic insomnia	Randomized, double-blind, placebo-controlled pilot study	A reduction in depressive symptoms and fatigue, along with decreased cortical excitability	[127]
Probiotics/Psychobiotics	*Bifidobacterium longum* NCC3001	Adults with mild to moderate anxiety and/or depressive symptoms	Randomized, double-blind, placebo-controlled pilot study	A reduction in depressive symptoms according to the HAD-D scale, an improvement in the physical component of quality of life, and decreased activation of the amygdala and fronto-limbic regions in response to threatening stimuli	[128]
Prebiotics	Galactooligosaccharides (GOS)	Healthy women aged 17–25 years	Randomized, double-blind, placebo-controlled trial	Neurochemical alterations and a transient increase in the abundance of *Bifidobacterium* within the gut microbiota	[99]
Prebiotics	Inulin	Students aged 18–23 years	Double-blind randomized controlled trial	Improvement in executive functions	[129]
Prebiotics	Polydextrose + GOS	Healthy full-term infants	Double-blind randomized controlled trial	Faster consolidation of daytime wakefulness	[130]
Polyphenols	Blackcurrant-based beverage 151 mg anthocyanins, 308 mg total polyphenols/day; also 150 mg of *Pinus radiata* extract (proanthocyanidins) and 200 mg L-theanine	Healthy women aged 18–45 years	Randomized double-blind placebo-controlled crossover trial	Improvement in working memory under multitasking conditions, along with reductions in tension/anxiety and irritability	[131]
Polyphenols	Cranberry drink: (~442 mg polyphenols)	Healthy students	Randomized double-blind placebo-controlled parallel-group trial	Improvement in short-term memory by week 12	[132]
Omega-3 fatty acids	DHA, EPA	Children aged 2–6 years with Autism Spectrum Disorder	Double-blind, randomized, placebo-controlled trial	A reduction in emotional symptoms, behavioral problems, and impact on quality of life (within the autism spectrum range)	[133]

HDRS—Hamilton Depression Rating Scale; MADRS—Montgomery-Asberg Depression Rating Scale; BRPS—Bainbridge-Ropers syndrome; GSRS—Gastrointestinal Symptom Rating Scale; PSQI—Pittsburgh Sleep Quality Index; AIS—Athens Insomnia Scale; ADHD—Attention Deficit Hyperactivity Disorder; CPRS-R-L—Conners’ Parent Rating Scale-Revised; CBCL—Child Behavior Checklist; CPT—Continuous Performance Test.

**Table 4 nutrients-17-03068-t004:** Characteristic changes in the intestinal microbiota in Parkinson’s disease and their pathogenetic consequences.

Microbiota Alterations	Associated Taxa	Functional Implication	Clinical/Pathogenetic Correlations	References
Changes in β-Diversity and Bacterial Families	↑ *Lactobacillaceae*, *Barnesiellaceae*, *Enterococcaceae*	Compromised intestinal barrier and increased systemic inflammation, promoting microglial activation and neurodegeneration	Associated with neurotransmitter imbalance and dysregulation of SCFA metabolism, collectively contributing to disease development and progression via gut–brain axis mechanisms	[157]
Increased α-Diversity	↓ *Lactobacillus*, *Sediminibacterium* ↑ *Clostridium* IV, *Aquabacterium*, *Holdemania*, *Sphingomonas*, *Clostridium* XVIII, *Butyricicoccus*, *Anaerotruncus*	Reduced metabolic activity of pathways involved in the synthesis of vitamins and cofactors, alongside enhanced energy metabolism, potentially contributing to neuroinflammation	The identified alterations in gut microbiota composition in Parkinson’s disease are pathogenetically correlated with disease duration, cognitive decline, depressive symptoms, and the presence of motor complications	[158]
Increased α-Diversity	↓ *Roseburia intestinalis*, *Faecalibacterium prausnitzii* ↑ *Akkermansia muciniphila*	Reduced production of SCFAs and polyamines due to impaired microbial synthesis of riboflavin and biotin	Thinning of the intestinal mucosa, increased epithelial permeability, enhanced neuroinflammation, and formation of pathological α-synuclein fibrils in the enteric nervous system	[159]
Reduced α-Diversity	↓ *Dorea*, *Bacteroides*, *Prevotella*, *Faecalibacterium*, *Bacteroides massiliensis*, *Stoquefichus massiliensis*, *Bacteroides coprocola*, *Blautia glucerasea*, *Dorea longicatena*, *Bacteroides dorei*, *Bacteroides plebeus*, *Prevotella copri*, *Coprococcus eutactus*, *Ruminococcus callidus*↑ *Christensenella*, *Catabacter*, *Lactobacillus*, *Oscillospira*, *Bifidobacterium*, *Christensenella minuta*, *Catabacter hongkongensis*, *Lactobacillus mucosae*, *Ruminococcus bromii*, *Papillibacter cinnamivorans*	Dysbiotic shifts in bacterial taxa are associated with chronic intestinal inflammation, metabolic disturbances, and potentially accelerated neurodegenerative processes	The observed alterations in microbial composition may trigger localized intestinal inflammation, subsequently promoting α-synuclein aggregation and the formation of Lewy bodies	[160]
Increased α- and β-Diversity	↓ *Faecalibacterium*, *Blautia*, *Fusicatenibacter* ↑ *Bacteroides*, *Corynebacteria*, *Deltaproteobacteria*, *Butyricimonas*, *Robinsoniella*, *Flavonifractor*	Enhanced degradation of the intestinal mucus layer, increased gut permeability, and the development of systemic inflammation	The observed gut microbiota alterations in Parkinson’s disease are associated with impaired intestinal barrier function, increased exposure to endotoxins and oxidative stress, and accumulation of α-synuclein in both the enteric and central nervous systems	[161]
Selective Dysbiosis with an Increase in Opportunistic Genera	↓ *Faecalibacterium* ↑ *Alistipes*, *Rikenellaceae*, *Bifidobacterium*, *Parabacteroides*	Decreased concentrations of branched-chain (BCAA) and aromatic amino acids	Disrupted metabolic activity of the microbial community, potentially affecting immune regulation and promoting neurodegenerative progression due to a deficit of metabolites involved in neuromodulation and maintenance of neuronal function	[162]
Reduction in SCFA-Producing Taxa	↓ *Lachnospiraceae*, *Coriobacteriaceae*, *Faecalibacterium*, *Fusicatenibacter*, *Roseburia*, *Blautia* ↑ *Clostridia* UCG014, *Christensenella*, *Oscillospiraceae*	Decreased SCFA synthesis and increased intestinal barrier permeability	The identified alterations in gut microbiota in Parkinson’s disease patients show clear pathogenetic correlations with key clinical symptoms, including functional constipation, neuroinflammation, metabolic disturbances, and progressive motor impairment	[163]
Reduction in SCFA-Producing Taxa	↓ *Blautia*, *Coprococccus*, *Roseburia*, *Faecalibacterium* ↑ *Proteobacteria*	Decreased SCFA synthesis, increased intestinal barrier permeability, and immune activation	The observed gut microbiota alterations in Parkinson’s disease patients are associated with inflammatory processes, oxidative stress, and α-synuclein aggregation	[164]

↑—increase; ↓—decrease.

**Table 5 nutrients-17-03068-t005:** Potential biomarkers for precision therapy of neurodegenerative diseases.

Biomarkers	Function	Impact on Nutrition	Association with Neurodegeneration	Therapeutic and Preventive Significance	References
MTHFR	Methylenetetrahydrofolate reductase	Folate metabolism	Disruption of synaptic transmission, epigenetic dysregulation, and diminished neurotrophic support are key contributors to the early pathogenesis of neurodegenerative disorders	An adequate folate status can prevent early epigenetic and metabolic disturbances, thereby reducing the risk of cognitive impairment	[211]
APOE	Apolipoprotein E	Microgliosis and diet sensitivity	The APOE4 allele promotes neurodegeneration by mediating microglial activation and amplifying neuroinflammatory processes	It serves as a therapeutic target and highlights the potential for preventive interventions through the modulation of diet and inflammatory responses	[212]
SNCA	α-synuclein	Eating disorder	The formation of pathogenic protein aggregates and their interaction with synaptic components, along with structural abnormalities caused by mutations, are directly linked to the mechanisms of neurodegeneration in Parkinson’s disease	It represents a key target for therapeutic development aimed at inhibiting alpha-synuclein aggregation, and forms the basis for preventive approaches focused on early-stage intervention in populations at risk for neurodegenerative diseases, such as Parkinson’s disease	[213]
MAPT	Tau protein	Regulation of nutrient transport in neurons	Pathological hyperphosphorylation and aggregation of tau protein disrupt axonal transport and contribute to neuronal degeneration	Modulation of MAPT gene expression and suppression of pathological tau protein synthesis represent a promising approach for the therapy and prevention of tauopathies, including Alzheimer’s disease and frontotemporal dementia, aiming to slow neurodegenerative processes and preserve cognitive function	[214]
LRRK2	Leucine-rich repeat kinase 2	Eating disorder	Mutations in the LRRK2 gene disrupt key cellular processes, including mitochondrial dysfunction, impaired autophagy, cytoskeletal dysregulation, and altered synaptic transmission, ultimately driving the progressive degeneration of dopaminergic neurons	The potential for targeted inhibition of kinase and GTPase activity offers a promising strategy to slow neurodegenerative processes and reduce the risk of Parkinson’s disease, particularly in carriers of pathogenic mutations	[215]
SOD1	Superoxide dismutase 1	Oxidative stress, metabolic imbalance, and neuromuscular degeneration	Mutations in the SOD1 gene are a known cause of familial amyotrophic lateral sclerosis (ALS), leading to selective degeneration of motor neurons in the brain and spinal cord	The targeted suppression of mutant SOD1 gene expression holds therapeutic potential for slowing neurodegeneration, preserving motor neuron function, and improving nutritional status	[216]
PARK7/DJ-1	Parkinson’s disease 7	Regulation of metabolism, gut microbiota, and mucosal barrier function	The PARK7/DJ-1 protein exerts neuroprotective effects through its antioxidant activity, inhibition of α-synuclein aggregation, prevention of protein glycation, regulation of neuroinflammation, and maintenance of BBB integrity under conditions of systemic inflammation	Antioxidant activity, prevention of glycation and aggregation of neurotoxic proteins, modulation of inflammatory responses, and maintenance of BBB integrity	[217]
PSEN1	Presenilin 1	Eating disorder	Mutations in the PSEN1 gene contribute to the development of neuroinflammation by promoting the activation of proinflammatory cytokines and disrupting the regulation of Notch signaling, thereby exacerbating neurodegenerative processes	The ability to identify carriers at high risk of neurodegeneration at an early stage is crucial for implementing preventive interventions and developing targeted therapies	[218]

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
