# Peer review of "Precision Nutrition and Gut–Brain Axis Modulation in the Prevention of Neurodegenerative Diseases"

_nutrients, 2025, doi:10.3390/nu17193068_

Round 1
Reviewer 1 Report
Comments and Suggestions for Authors
The manuscript is well-structured, scientifically sound, and provides an exciting insight into the neuroprotective potential of chrononutrition and personalized nutrition. Although clinical evidence remains limited, the work is inspiring and offers valuable guidance for future research and applications. To further enhance readability and practical relevance, the inclusion of some practical recommendations is suggested. It would also be beneficial to place a glossary of abbreviations beneath the tables and to briefly describe the research methodology following the introduction. The references are relevant, the manuscript is well-edited and thorough, and the scientific language is appropriate. I recommend this manuscript for publication.
Author Response
Dear Reviewer,
We would like to express our sincere gratitude for your thorough evaluation of our manuscript and for the valuable comments that have significantly improved the quality of the presented work.
In response to your recommendations, we have made the following revisions to the text:
– Added Section “2. Methods” — following the introduction, we provide a concise description of the applied research methodology, which offers the reader a clearer understanding of the scientific approach and the logic of the material presentation.
– Glossary of abbreviations placed after the tables — this improves readability and facilitates comprehension of the presented information.
– Expanded review of regulatory and expert positions — the text now incorporates the perspectives of such authoritative organizations as EFSA, ESPEN, ISAPP, and WHO regarding the use of functional foods and probiotics in the prevention of neurodegenerative diseases.
– Inclusion of practical recommendations — to enhance the applied value of the study, we have added several suggestions based on current scientific evidence and international practices.
We are grateful for your positive assessment of the manuscript’s structure, scientific rigor, and academic style, as well as for your remark on the importance of practical orientation. We are confident that the implemented revisions have strengthened the value of the article and made it more useful for both researchers and professionals in the field of nutrition and neuroprotection.
We once again express our gratitude for your constructive comments and your favorable recommendation for publication.
Reviewer 2 Report
Comments and Suggestions for Authors
The present review article, authored by Dilyar Tuigunov and colleagues explains about the “Precision Nutrition and Gut-Brain Axis Modulation in the Prevention of Neurodegenerative Diseases”., The review highlights the gut microbiota as a pivotal target for the development of precision-based dietary strategies in the prevention and migration of neurodegenerative disorders. Mainly focused to key bioactive components such as prebiotics, probiotics, psychobiotics, dietary fiber, omega-3 fatty acids, and polyphenols that critically participate in regulating the gut-brain axis. The findings presented in the review emphasize the potential of integrating precision nutrition with targeted modulation of the gut-brain axis as a multifaceted approach to reducing the risk of neurodegenerative diseases and preserving cognitive health. The author’s conclusions are, precision nutrition acts as a promising and scientifically substantiated strategy focused on the personalized composition of the microbiota, metabolic profile, epigenetic markers and nutritional needs.
In general, the review is interesting, important and potentially helpful for scientific communities and clinical understanding. Especially with the fact that aging populations with neurodegenerative diseases are increasing, unfortunately available interventions and treatments for AD and other neurodegenerative diseases are highly expensive and not cost effective. Dietary changes and microbiota are so important to slow/ prevent the diseases progression.
Manuscript can be accepted after following minor revisions.
- On page #2 line 77, please correct the spelling of “amyloid plagues” to -plaques.
- The review is too lengthy and somewhat minimizes the readability, can be shortened e.g., On page #3, line124-161can be, too much detail about vagus nerve.
- Please remove “I” from line 204, page#4, or if author mean to say “In”, either way, please correct.
- On page 6, line#291, please change “carrier” to barrier (including modulation of the integrity of the blood brain carrier - 290).
- 1, in Gut microbiota, some captions are hard to read, if possible, please replace with better resolution figure.
- Word should be abbreviated when appeared first in the manuscript than abbreviation can be used throughout the manuscript as needed, (e.g., On page 8, line 337, (blood-brain barrier (BBB)).
- Line# 380-381-grammer.
- Line 400, please correct (observed int eh oral microbiome of patients).
- Line 505, please replace “if” with is (“which if due to their”).
- For the limitations and the future challenges from line 971-1011, reviewer is somewhat agreeing with the author, but much progress has been made in this area including microbiome analysis (e.g., Reviewed in PMID: 37128855), and NGS has revolutionized scalability, speed and cost effectiveness to perform a wide range of studies, including the analysis of microbial communities associated with host and environment.
Author Response
Dear Reviewer,
We sincerely thank you for your thorough evaluation of our manuscript and for the valuable recommendations that have enhanced its quality and scientific significance.
In accordance with your comments, the following revisions have been made to the text:
Corrections of typos and grammatical errors:
– “amyloid plagues” was corrected to “plaques”;
– the redundant symbol “I” was removed;
– “carrier” was corrected to “BBB”;
– “observed int eh oral microbiome of patients” → “observed in the oral microbiome of patients”;
– “if” was corrected to “is.”
– Optimization of text structure:
– To improve readability, excessively detailed fragments were shortened, in particular the description of the vagus nerve (page 3, lines 124–161).
– Quality of illustrations:
In the section on microbiota, the figure was added in high resolution (PNG format).
– Use of abbreviations:
Abbreviations are now provided after the first mention of terms (e.g., “blood–brain barrier (BBB)”), in line with academic standards.
– Section “Limitations and Future Challenges”:
Revised in light of your comment on advances in microbiome analysis. We included a reference to the modern potential of next-generation sequencing (NGS) technologies, which substantially enhance scalability, speed, and cost-effectiveness of research (see, e.g., PMID: 37128855).
We are grateful for your positive assessment of the manuscript, as well as for the recommendations that helped us to strengthen its scientific and practical value. We are confident that the revisions have made the article more coherent, readable, and brought it in line with the high standards of the journal.
Once again, we thank you for your attention and for your recommendation for publication.
Reviewer 3 Report
Comments and Suggestions for Authors
The submitted manuscript represents an ambitious attempt to synthesize contemporary knowledge on the influence of gut microbiota, its metabolites, and interactions with the gut-brain axis on the development of neurodegenerative diseases, with particular emphasis on the role of precision nutrition. The authors address a wide range of topics – from neurophysiology, through immunology and metabolism, to the applications of nutrigenomics and chrononutrition.
Despite the broad scope of the topic, the article requires significant improvements in terms of structural clarity, data integration, critical literature review, and linguistic accuracy. Detailed comments are provided below.
1. Lack of coherent narrative and synthesis
The article resembles a comprehensive compilation of facts rather than a synthetic analysis with a clear direction.
Please develop a conceptual framework around which the individual sections will be organized. This could be done, for example, by type of dietary intervention (probiotics, fiber, PUFAs), mechanism of action (SCFAs, cytokines, neurotransmitters), or disease (AD, PD, MS).
In the Conclusions section (pp. 27–28), you should critically evaluate the presented data and propose coherent recommendations for the future.
2. Overly technical descriptions of neuroanatomy (section 2.1)
The description of the structure of the vagus nerve, its fibers, nuclei, etc. (pp. 3–4) is excessively extensive and inadequate for the purpose of a review. This section should be shortened and simplified, emphasizing the functional importance of the vagus nerve in the gut-brain axis.
3. Failure to distinguish between preclinical and clinical evidence
In many places, the results of in vitro, animal, and human studies are summarized without clearly indicating their source. Example: sections 3.1–3.3.
Please separate or clearly label the results of clinical and experimental studies – this is crucial for assessing the validity of the evidence.
4. The section on prebiotics (4.1) is based primarily on animal models.
Conclusions on the effects of GOS or 2′-FL on cognitive function (pp. 15–16) are based on studies in rats and mice.
It would be advisable to add several randomized clinical trials involving humans.
5. The section on probiotics and psychobiotics (4.2) requires updating.
The paper does not mention specific strains considered psychobiotics (e.g., Lactobacillus helveticus R0052, Bifidobacterium longum R0175) that have clinical data.
There is also no reference to meta-analyses assessing the effectiveness of probiotics in neurocognitive disorders.
6. The "Limitations" section (Chapter 6) is too short.
It should be expanded – for example, to address the challenges of personalizing nutritional interventions, the variability of microbiota composition, difficulties in selecting probiotics with CNS activity, population heterogeneity, etc.
7. A diagram or table presenting the precision nutrition model is missing.
It would be helpful to add a clear diagram summarizing the precision nutrition approach: input data (genome, microbiome, phenotype), analysis (AI, omics), and output (dietary interventions).
8. Many passages require linguistic corrections.
Examples:
"...may lead to the prevention and migration of neurodegenerative disorders." → "mitigation," not "migration."
"...reduction of proinflammatory cytokine levels..." – incorrect syntax.
I recommend proofreading by a native speaker.
9. Lack of a clear summary of clinical data (e.g., in table form)
Please consider adding a table summarizing, for example, the effects of individual groups of bioactive ingredients (probiotics, prebiotics, polyphenols, PUFAs) in clinical trials – cognitive parameters, inflammatory markers, biomarkers of neurodegeneration, etc.
10. Missing references to current clinical recommendations
A reference to the EFSA, ESPEN, ISAPP, or WHO positions regarding the use of functional foods and probiotics in the prevention of neurodegenerative diseases would be useful.
Author Response
Dear Reviewer,
We sincerely thank you for your in-depth analysis of our manuscript and for the constructive recommendations that have substantially enhanced its scientific and practical quality.
- Conceptual structure and coherent narrative
In line with your suggestion, the overall logic of the paper has been revised. The manuscript now presents a coherent narrative centered on the concept of precision nutrition and modulation of the gut–brain axis as a key strategy for the prevention of neurodegenerative diseases. The argument unfolds by addressing the global burden of dementia, mechanisms of microbiota–CNS interactions, the role of dietary interventions (probiotics, prebiotics, polyphenols, omega-3), and the prospects of nutrigenomics and personalized approaches for preserving cognitive health.
- Condensed description of neuroanatomy
Section 2.1 has been revised: overly technical details on the vagus nerve anatomy were removed, while emphasis was placed on its functional role within the gut–brain axis.
- Separation of preclinical and clinical data
All findings were reviewed and clearly classified according to source type, presented in the sequence in vitro, in vivo and clinical trials. Relevant labels have been provided in the text to improve the transparency of the evidence base.
- Section on prebiotics (4.1)
Data from several randomized clinical trials were added regarding the effects of GOS and FOS on cognitive functions, with clear differentiation between animal and human findings.
- Section on probiotics and psychobiotics (4.2)
Updated with recent evidence: information on specific psychobiotic strains (e.g., Lactobacillus helveticus R0052, Bifidobacterium longum R0175) was included, along with references to recent meta-analyses assessing their efficacy in neurocognitive disorders.
- Section “Limitations” (Chapter 6)
Substantially expanded: challenges of dietary personalization, high variability in microbiota composition, difficulties in selecting probiotic strains with CNS-related activity, and heterogeneity of study populations are now addressed.
- Addition of illustrative materials
A schematic figure illustrating the precision nutrition model has been prepared (Figure 3).
- Language corrections
Several inaccuracies were corrected:
– “migration” replaced with “mitigation”;
– syntactic refinements, including the expression “reduction of proinflammatory cytokine levels”;
– additional language editing was carried out to improve the quality of academic English.
- Summary table of clinical data
A new table was added to Section 4 (Dietary Interventions as Modulators of Gut–Brain Axis Mechanisms), summarizing the results of clinical studies across groups of bioactive ingredients (probiotics, prebiotics, PUFAs, polyphenols), with reference to cognitive parameters, inflammatory markers, and neurodegeneration biomarkers.
- References to current clinical guidelines
The text was supplemented with positions of EFSA, ESPEN, ISAPP, and WHO concerning the use of functional foods and probiotics in the prevention of neurodegenerative diseases.
We appreciate your recommendations, which have helped us make the manuscript more structured, analytically consistent, and practice-oriented. All revisions have been highlighted in color for ease of review. These improvements have strengthened the scientific value of the review and enhanced its relevance for both researchers and clinicians.
Once again, we sincerely thank you for your thoughtful review and for the recommendations that allowed us to substantially improve the manuscript.
Round 2
Reviewer 3 Report
Comments and Suggestions for Authors
The Authors have revised manuscript accordingly to Reviewer's suggestions.